# DAIL: Beyond Task Ambiguity for Language-Conditioned Reinforcement Learning

**Runpeng Xie**[*1], **Quanwei Wang**[*2], **Hao Hu**[3], **Zherui Zhou**[4], **Ni Mu**[2], **Xiyun Li**[5], **Yiqin Yang**[†1],
**Shuang Xu**[1], **Qianchuan Zhao**[2], **Bo Xu**[†1]

[1]The Key Laboratory of Cognition and Decision Intelligence for Complex Systems,
Institute of Automation, Chinese Academy of Sciences, Beijing, China
[2]Department of Automation, Tsinghua University    [3]Moonshot AI
[4]Department of Computer Science and Engineering, Washington University    [5]Tecent AI Lab
xierunpeng2021@ia.ac.cn, wqw21@mails.tsinghua.edu.cn

## Abstract

Comprehending natural language and following human instructions are critical capabilities for intelligent agents. However, the flexibility of linguistic instructions induces substantial ambiguity across language-conditioned tasks, severely degrading algorithmic performance. To address these limitations, we present a novel method named DAIL (Distributional Aligned Learning), featuring two key components: distributional policy and semantic alignment. Specifically, we provide theoretical results that the value distribution estimation mechanism enhances task differentiability. Meanwhile, the semantic alignment module captures the correspondence between trajectories and linguistic instructions. Extensive experimental results on both structured and visual observation benchmarks demonstrate that DAIL effectively resolves instruction ambiguities, achieving superior performance to baseline methods. Our implementation is available at https://github.com/RunpengXie/Distributional-Aligned-Learning.

## 1 Introduction

Artificial agents are anticipated to master diverse skills while effectively interpreting human instructions and generalizing across various tasks. Therefore, comprehension and following of natural language emerges as critical capabilities for agents in this context. For example, language-conditioned agents have achieved remarkable success in robotic manipulation [36, 6], text-based environments [35, 7], visual navigation [62, 22], and autonomous driving [14, 51]. The fundamental requirement has propelled language-conditioned reinforcement learning (RL) to the forefront of research, which focuses on enabling agents to interpret and execute natural language instructions through RL frameworks.

Recent advancements in the language-conditioned RL domain have focused on bridging the gap between linguistic understanding and decision-making processes, aiming to create agents capable of executing complex instructions with human-like adaptability. For example, some works [12, 4] integrate language-conditioned policy with trial-and-error learning, significantly improving the performance and sample efficiency in robot task acquisition. Meanwhile, some studies [18, 19] leverage expert demonstrations to map language instructions to reward signals directly to address the issue of sparse rewards in language-conditioned RL. However, linguistic instructions exhibit high flexibility, which induces exponential growth in task space. In this case, identical tasks may have

---

*Equal contribution.
†Correspondence to Yiqin Yang and Bo Xu.

39th Conference on Neural Information Processing Systems (NeurIPS 2025).

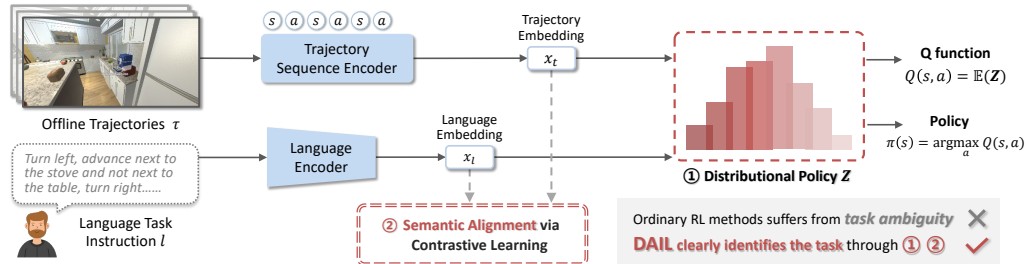

Figure 1: The framework of our method. The key ideas are: (1) use the distributional language-guided policy to aid task discrimination, (2) use the trajectory-wise semantic alignment module to extract task representation.

divergent expressions, while distinct tasks share overlapping language instructions. This variability makes language-conditioned RL methods face the significant challenge of task ambiguity, which hinders the agent from discerning the connection between rewards and task objectives, thereby significantly impairing learning efficiency.

To solve this issue, we propose a novel method, DAIL, which consists of two main components: a distributional language-guided policy and a trajectory-wise semantic alignment module. Specifically, the distributional policy module [3] estimates the value distribution, preserving more information to aid task discrimination. Theoretically, we analyze the sample complexity required to guarantee task disambiguation and establish that distributional estimation methods are sample-efficient in offline settings. On the other hand, the semantic alignment module constrains the language instruction representations by maximizing the mutual information between trajectories and the instructions, thereby achieving better differentiation across instructions. With theoretical guarantees, the two modules enable precise disambiguation and execution of linguistic instructions, thereby improving learning efficiency.

We conduct extensive experiments on both structured observation [10] and visual observation [54] benchmarks. This design is to validate the external validity of the DAIL agent, progressing from less complex structured inputs to more complex and expressive visual observations. The experimental results show that DAIL outperforms the state-of-the-art language-conditioned RL methods in both benchmarks. Further, the visualization analysis demonstrates that DAIL can learn a non-ambiguous task representation compared with baselines. Our main contributions are summarized as follows:

- First, we highlight the critical issue of task ambiguity and empirically analyze the limitations of current mainstream methods. We define the task distinction in our setting and analyze the sample complexity to avoid task ambiguity theoretically.

- Second, we propose DAIL, a simple yet efficient language-conditioned learning framework, which addresses the task ambiguity issue based on distributional policy and semantic alignment.

- Lastly, we conduct extensive experiments to show that DAIL significantly outperforms conventional language-conditioned methods. The results indicate that by improving task discrimination, we can effectively mitigate the task ambiguity issue, thereby broadening the application of language-conditioned RL.

## 2 Preliminaries

**Language-conditioned RL**  Based on contextual markov decision process (CMDP) [20], we consider Language-conditioned Markov Decision Process (LCMDP) as a model consisting of a tuple $\mathcal{M} = (\mathcal{S}, \mathcal{A}, P, r, \gamma, \mathcal{L}, p_0, p_l)$, where $\mathcal{S}$ denotes the state space, $\mathcal{A}$ represents the action space, $\mathcal{L}$ is the language instruction space, $P(s'|s, a, l)$ represents the probabilistic transition model, $r : \mathcal{S} \times \mathcal{A} \times \mathcal{L} \to \mathbb{R}$ is the reward function conditioned on language instructions and $\gamma$ is the discount factor. $p_0$ represents the probability distribution of the initial state, and $p_l$ denotes the probability distribution of language instruction. We establish language instructions $l$ as task descriptors, and each instruction uniquely specifies a task.

Language-conditioned RL aims to obtain a policy $\pi(\cdot|s,l)$ that maximizes the cumulative discounted returns under a specific distribution of language instructions:

$$\pi^* = \arg\max_\pi \mathbb{E}_\pi \left[\sum_{t=0}^\infty \gamma^t r(s_t, a_t, l)\right], \tag{1}$$

where $s_0 \sim p_0(\cdot), l \sim p_l(\cdot), a_t \sim \pi(\cdot|s_t, l)$ and $s_{t+1} \sim P(\cdot|s_t, a_t, l)$. The temporal difference loss in language-conditioned RL is adapted as follows:

$$L_{\text{TD}}(\theta) = \mathbb{E}_{(s,a,s',l)\sim\mathcal{D}}[(r(s,a,l) + \gamma\max_{a'} Q_{\hat{\theta}}(s', a', l) - Q_\theta(s, a, l))^2] \tag{2}$$

where $Q_\theta(s, a, l)$ is the parameterized $Q$-function conditioned language instruction $l$, and $Q_{\hat{\theta}}(s, a, l)$ is the target $Q$-network.

**Offline RL**   Due to the high cost of real-time interaction with the environment, we consider the offline learning setting, in which we learns a policy $\pi$ without interacting with an environment. Rather, the learning is based on a dataset $\mathcal{D}$ generated by a behavior policy $\pi_\beta$. One of the major challenges in offline RL is the issue of distributional shift [17], where the learned policy is different from the behavioral policy. Existing offline RL methods apply various forms of regularization to limit the deviation of the current learned policy:

$$\pi^* = \arg\max_\pi [J_\mathcal{D}(\pi) - \alpha D(\pi, \pi_\beta)], \tag{3}$$

where $J_\mathcal{D}(\pi)$ is the cumulative discounted return of policy $\pi$ on the empirical MDP induced by the dataset $\mathcal{D}$, and $D(\pi, \pi_\beta)$ is a divergence measure between $\pi$ and $\pi_\beta$. As for the language-conditioned task, we will provide the language instruction in the evaluation. To make our writing more concise, let $\tau$ be a full trajectory, and $\tau_t$ be the trajectory ending at time-step $t$. Let $p_\mathcal{D}(\tau, l)$ represents the joint distribution of trajectory-instruction pairs in the dataset $\mathcal{D}$.

## 3   Ambiguity on Language-Conditioned Tasks

In practical tasks, language instructions have high flexibility. For example, similar tasks may employ divergent expressions, while distinct tasks share overlapping language instructions. The variability induces exponential growth of the task space. Therefore, when the number of language instructions increases, the agent is required to accurately identify the tasks; otherwise, it will significantly affect the agent's performance. We name this issue the ambiguity in language-conditioned tasks. We give a formal definition of semantics distinction from instructions as follows.

**Definition 1** (Semantics Instructions Distinction). *In a multi-task RL setting with known task instruction space $\mathcal{L}$ and unknown semantics space $\mathcal{G}$, for a task distinction threshold $\delta$ and sub-optimality gap $\epsilon$, two task instructions $l_i, l_j \in \mathcal{L}$ are considered with **different underlying semantics**, $g_i \not\leftrightarrow g_j$, if the expected Q-values under any shared $\epsilon$-optimal policy $\pi$ satisfy:*

$$\mathbb{E}_\pi [|Q_\pi(s, a, l_i) - Q_\pi(s, a, l_j)|] \geq \delta.$$

*Conversely, if*

$$\mathbb{E}_\pi [|Q_\pi(s, a, l_i) - Q_\pi(s, a, l_j)|] \leq \delta/2,$$

*then $l_i$ and $l_j$ are considered with the **same underlying semantics**, $g_i \leftrightarrow g_j$, where $s_0 \sim p_0(\cdot), a \sim \pi(\cdot|s), s' \sim p(\cdot|s, a)$. $V_\pi(s) \geq V^*(s) - \epsilon, \forall s \in \mathcal{S}$, $V^*$ is optimal value function.*

To make this point more straightforward, we introduce a toy experiment based on the Minigrid environment [11]. The setup consists of 10 accessible goal positions $\mathcal{G} = \{g_0, g_1, ..., g_9\}$, where $\mathcal{G}$ represents the set of all possible goal positions. The agent (red triangle-shaped) must follow a given instruction $l \in \mathcal{L}$ to navigate to a specific goal position (Left of Figure 2). We simulate instructions using numerical task IDs, making $\mathcal{L} \subset \mathbb{N}$. We employ a random mapping $F : \mathcal{L} \rightarrow \mathcal{G}$ to assign each $l$ to goal position $g$, which simulates semantics of instructions and is hidden from the agent (Middle of Figure 2). With these settings, we can control the number of instructions $|\mathcal{L}|$ by simply adjusting the set of valid task IDs and the mappings. We conduct 10 experiments, varying the number of instructions from 1 to $2^9$. For each experiment, we generate a new mapping $F$ and collect 1024 random trajectories as the offline dataset.

We evaluate the standard offline RL algorithm, CQL [30], on the above settings. In addition, to test how model size affects the task ambiguity, we create an enhanced version called CQL-double by

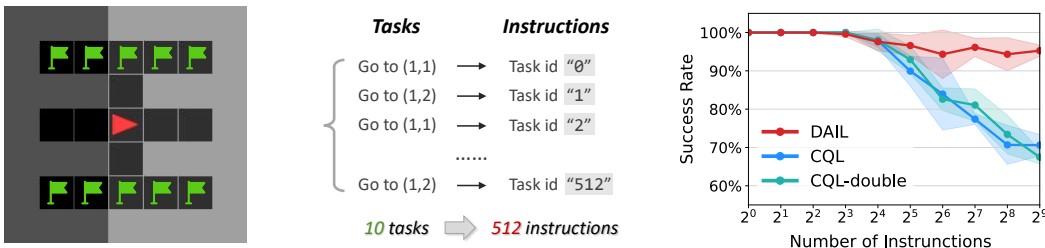

Figure 2: **Left:** The green flags in the map denote the accessible goals. **Middle:** An illustration of the mapping between goal positions and instructions. **Right:** Average success rates over 100 evaluations for each number of instructions and 3 seeds.

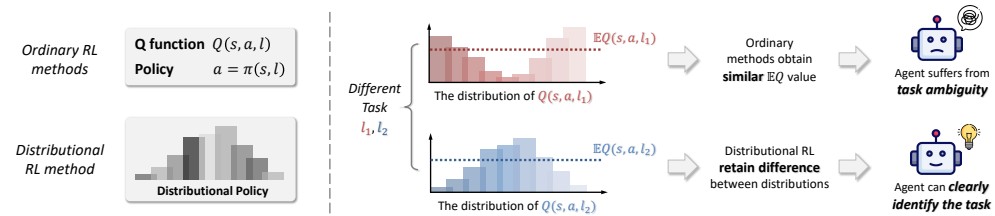

Figure 3: An illustrative case where $Q(s, a)$ functions in two tasks share the same expectations but have different distributions. In this case, traditional RL cannot discriminate between the tasks while distributional RL can. Please refer to Appendix D for more details.

doubling its parameters. The experimental results in Figure 2 show that if the number of instructions is small (e.g., under 16), agents can understand instructions and reach the goal position with high success. However, as instructions multiply, the number of demonstrations for each instruction declines, which increases task ambiguity and consequently makes CQL and CQL-double performance drop significantly. These observations reveal fundamental limitations in language-conditioned RL regarding task ambiguity, highlighting that model scaling alone remains insufficient to resolve this challenge. For detailed information on the toy experiment, please refer to Appendix E.

## 4 Method

To address the issue of task ambiguity, we hope to improve the algorithm's ability to distinguish tasks. In this section, we propose the Distributional Aligned Learning algorithm (DAIL), which enhances the task differentiability from policy and representation. Specifically, DAIL adopts the distributional language-guided policy to estimate the value distribution, preserving more information to aid task discrimination. On the other hand, DAIL uses the trajectory-wise semantic alignment module to extract task representations and help discriminate different instructions by maximizing the mutual information between trajectories and instructions. We show the overall framework of DAIL in Figure 1 and Algorithm 1 in Appendix B.

### 4.1 Distributional Language-Guided Policy

Current RL methods aim to estimate the expectation of the cumulative discounted reward. However, as shown in Section 3, when the number of instructions increases while keeping the number of actually-distinct tasks constant, the estimated expectation for different task instructions becomes similar. As a result, it is difficult for the agent to complete the task instructions accurately. Differently, the distributional technique addresses this issue by calculating the distribution of the cumulative discounted reward, which is more distinguishable than expectation, as illustrated through an extreme yet straightforward example in Figure 3. For a fixed policy $\pi$, let the random variable $Z^\pi$ represent

the cumulative discounted reward obtained along the policy $\pi$, and it has the following relationships:

$$V^\pi(s, l) := \mathbb{E}[Z^\pi(s, l)] = \mathbb{E}\left[\sum_{t=0}^{\infty} \gamma^t r(s_t, a_t, l)|s_0 = s, \pi\right]$$

$$Q^\pi(s, a, l) := \mathbb{E}[Z^\pi(s, a, l)] = \mathbb{E}\left[\sum_{t=0}^{\infty} \gamma^t r(s_t, a_t, l)|s_0 = s, a_0 = a, \pi\right], \tag{4}$$

Let $\mathcal{Z}$ denote the space of value distributions. In the following, for simplicity of notation, we denote $Z^\pi \in \mathcal{Z}$ as $Z$. Instead of estimating the expectation, we calculate the probability distribution of the random variable $Z$:

$$\mathcal{T}^\pi Z(s, a, l) \overset{D}{:=} R(s, a, l) + \gamma Z(s', a', l), \tag{5}$$

where $\mathcal{T}$ is the distributional Bellman operator, and $A \overset{D}{:=} B$ denotes that $A$ equals $B$ by probability laws. $R \in \mathcal{Z}$ is a function depicting the reward distributions. Since modeling continuous distributions is challenging, we can discretize the value function distribution $Z$ and train it by minimizing the cross-entropy loss:

$$\mathcal{L}_{\text{Dist}}(\theta) = \mathbb{E}_{(\tau, l) \sim p_\mathcal{D}(\cdot, \cdot)} \frac{1}{T} \sum_{t=0}^{T-1} D_{\text{KL}}\left(\Phi \hat{\mathcal{T}} Z_{\hat\theta}(s_t, a_t, l) \| Z_\theta(s_t, a_t, l)\right), \tag{6}$$

where $T$ is the trajectory length, $\Phi\hat{\mathcal{T}}$ is the sample Bellman update, which projects the value function distribution onto the parametric discrete distribution. $Z_\theta$ and $Z_{\hat\theta}$ are estimated value distributions parameterized by $\theta$ and $\hat\theta$, respectively. By optimizing Equation 6, we obtain the distribution $Z_\theta$ that exhibits strong discriminative power across different instructions $l$. Please refer to Appendix C for the details.

In addition, we conduct the following theoretical analysis. Let $n_{\text{value}}, n_{\text{dist}}$ denote the number of samples needed to avoid task ambiguity for value-based and distributional settings, respectively. As shown in Theorem 1 and Corollary 2, distributional RL achieves better sample efficiency compared with estimating the expectation when the number of tasks is sufficiently large, $n_{\text{value}} \geq n_{\text{dist}}$. Proofs and further details can be found in Appendix D.

**Theorem 1** (Sample Complexity for Task Instruction Disambiguation)**.** *Consider an offline multi-task RL setting with $M$ distinct tasks (with different semantics). In direct Q-value estimate setting, suppose $Q(s, a, l) \in [0, Q_{\max}]$, $\forall(s, a, l) \sim \mathcal{D}$ with finite $Q_{\max}$, and the task distinction threshold is $\delta > 0$. When the number of training samples $n_{\text{value}}$ satisfies:*

$$n_{\text{value}} \geq \frac{C_{\text{value}} \log(3M^2/\eta)}{\delta^2}.$$

*The mean value estimate algorithm achieves task-level semantic disambiguation with confidence at least $1 - \eta$. In the distributional RL setting, let $Z(s, a, l)$ denote the learned return distribution, and suppose the task distinction threshold is given by a 1-Wasserstein distance $d > 0$. Then, to ensure semantic disambiguation of task instructions with confidence at least $1 - \eta$, it suffices that:*

$$n_{\text{dist}} \geq \frac{C_{\text{dist}} \log(3M^2/\eta)}{d^2},$$

*where $C_{\text{value}}, C_{\text{dist}} > 0$ are universal constants depending on certain attributes of Q-value distribution.*

## 4.2 Trajectory-Wise Semantic Alignment

In this subsection, we focus on learning trajectory embedding that enhances the correspondence between instructions and trajectories, thereby reducing task ambiguity at the representational level. To achieve this, we attempt to maximize the mutual information between the language instructions and trajectory:

$$w = \max_w I(X_\tau(w); X_l(w)), \tag{7}$$

where $w$ is the parameter of the representation module, $X_\tau, X_l$ are the random variables of trajectory embedding and language instruction, respectively. For the trajectory embedding $x_\tau$, we adopt the

sequence model. Specifically, we first use $u_w(\cdot, \cdot)$ to encode the state-action pairs, and then pass the sequence of embeddings through a sequence model $h_w(\cdot)$:

$$x_\tau = h_w(u_w(s_1, a_1), u_w(s_2, a_2), ..., u_w(s_T, a_T)). \tag{8}$$

As for the language instruction representation, we employ a language encoder to tokenize and encode the instructions into language embeddings $x_l$. Please refer to Appendix F for the detailed model architecture. Let $f_w(\tau, l) = \frac{x_\tau^T x_l}{||x_\tau|| \, ||x_l||}$ measure the similarity between the trajectory embedding and language instruction representation. Since minimizing InfoNCE is equivalent to maximizing the lower bound of the mutual information [45], we can maximize the mutual information in Equation 7 by minimizing the following NCE loss:

$$\mathcal{L}_c(w) = \mathbb{E}_{\substack{(\tau, l^+) \sim p_{\mathcal{D}}(\cdot, \cdot) \\ l^- \sim p_l(\cdot)}} [\log \sigma(f_w(\tau, l^+)) + \log(1 - \sigma(f_w(\tau, l^-)))], \tag{9}$$

where $l^+$ denotes the positive samples, which are sampled from the distribution of trajectory-instruction pair $p_{\mathcal{D}}(\cdot, \cdot)$. $l^-$ denotes the negative samples, generated by uniform sampling over the language instructions from the offline datasets.

### 4.3 Practical Implementation

To address the partial observability issue in practical implementation, the trajectory encoding $x_t$ defined in Equation 8 is incorporated as an additional input $(s_t, a_t, x_{t-1}, l)$. The Equation 5 is transformed into correspondingly:

$$\mathcal{T}^\pi Z(s_t, a_t, x_{t-1}, l) \overset{D}{:=} R(s_t, a_t, l) + \gamma Z(s_{t+1}, a_{t+1}, x_t, l). \tag{10}$$

In practice, we estimate the value distribution $Z_\theta$ with discrete distribution, using a set of atoms $\{z_i = V_{\text{MIN}} + i\Delta z\}_{i=1}^{M-1}, \Delta z = \frac{V_{\text{MAX}} - V_{\text{MIN}}}{M-1}$. $V_{\text{MIN}}, V_{\text{MAX}} \in \mathbb{R}$ are the lower and upper bounds of the distributions with support, respectively, and $M \in \mathbb{N}$ is the number of atoms. Then the discrete value distribution is modeled as:

$$Z_\theta(s_t, a_t, x_{t-1}, l) = z_i, \text{ with probability } p_i(s_t, a_t, x_{t-1}, l) := \frac{e^{\theta_i(s_t, a_t, x_{t-1}, l)}}{\sum_j e^{\theta_j(s_t, a_t, x_{t-1}, l)}} \tag{11}$$

where $\theta_i : \mathcal{S} \times \mathcal{A} \times \mathcal{X} \times \mathcal{L} \to \mathbb{R}$ is a parametric model employed to approximate $Z_\theta$ and updated by Equation 6. Based on the analysis in Section 4.1, we compute $Q$-function with distributional mechanism by $Q_\theta(s_t, a_t, x_{t-1}, l) = \mathbb{E}[Z_\theta(s_t, a_t, x_{t-1}, l)] = \sum_{i=0}^{M-1} p_i(s_t, a_t, x_{t-1}, l)z_i$. Please refer to Appendix C for the details.

In addition, we consider the offline learning setting in this work, which learns a policy without interacting with the environment. For this reason, we adopt the standard offline learning term, CQL($\mathcal{H}$) [30], to address the distribution shift issue in offline RL learning [17]:

$$\mathcal{L}_{\text{CQL}}(\theta) = \mathbb{E}_{(\tau, l) \sim p_{\mathcal{D}}(\cdot, \cdot)} \frac{1}{T} \sum_{t=0}^{T-1} \log \sum_a \exp(Q_\theta(s_t, a, x_{t-1}, l)) - \mathbb{E}_{a \sim \pi_\beta(a|s)}[Q_\theta(s_t, a, x_{t-1}, l)], \tag{12}$$

Combining all the above loss functions, the total loss function is:

$$\mathcal{L}_{\text{tot}} = \mathcal{L}_{\text{Dist}} + \lambda \mathcal{L}_c + \alpha \mathcal{L}_{\text{CQL}} \tag{13}$$

where $\lambda, \alpha$ are weights of the trajectory-wise semantic alignment module and the offline learning term, respectively. Finally, we select the action with the highest $Q$-value:

$$a_t^* = \underset{a}{\operatorname{argmax}} Q_\theta(s_t, a, x_{t-1}, l) \tag{14}$$

The complete process of our method is shown in Algorithm 1 in Appendix B and Figure 14 in Appendix F.

## 5 Experiments

We designed our experiments to answer the following questions: *Q1*: How does DAIL compare to other state-of-the-art methods on offline language-conditioned tasks? *Q2*: How does DAIL perform as the number of tasks explodes? *Q3*: Can DAIL learn a meaningful alignment between trajectories and instructions? *Q4*: What is the contribution of each of the proposed techniques in DAIL?

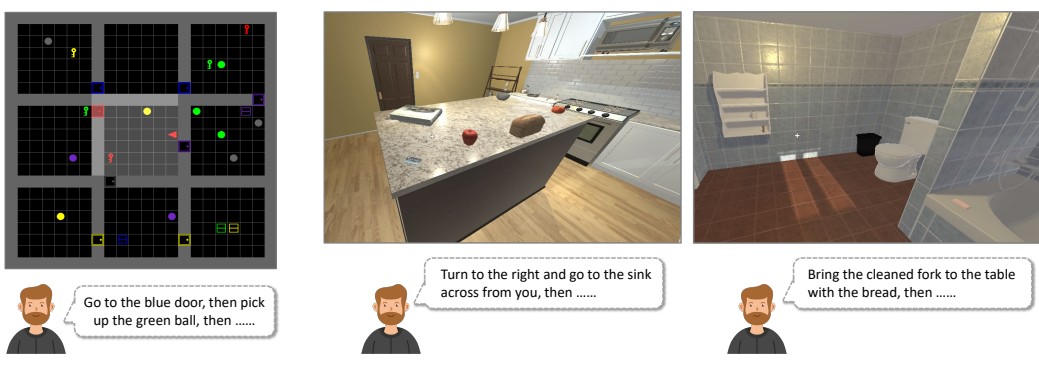

Figure 4: **Left:** An example of the `SynthLoc` task in the BabyAI environment: "put the yellow key next to a green ball". **Right:** Two example scenes with instructions from ALFRED. Agents are asked to finish specific tasks in the 3D household environment according to received instructions.

Table 1: Success rate of out-of-distribution BabyAI tasks. Each score is evaluated over 3 seeds.

| Tasks | GCBC | BC-Z | GRIF | IQL | CQL | DAIL (ours) |
|---|---|---|---|---|---|---|
| Open | 94.4±2.5 | 93.7±1.0 | 95.9±1.7 | 98.0±0.4 | 98.8±0.5 | **99.0±0.2** |
| Goto | 90.3±1.6 | 76.9±3.0 | 88.8±2.6 | 86.1±1.2 | 88.9±2.1 | **91.3±1.0** |
| PickUp | 78.4±2.1 | 45.4±1.5 | 75.6±3.9 | 70.4±3.6 | 71.9±2.2 | **87.6±2.0** |
| PutNext | 27.4±1.6 | 11.2±3.3 | 22.5±2.7 | 21.4±3.1 | 27.6±0.8 | **49.1±1.8** |
| All | 74.1±0.7 | 57.9±1.8 | 71.2±2.6 | 69.7±2.3 | 72.6±0.4 | **81.7±1.3** |

## 5.1 Experimental Setting

We evaluate various methods in language-conditioned tasks, with detailed introductions as follows:

**BabyAI**  [10] is a language learning research platform with different levels of tasks, shown on the Left of Figure 4. We choose level `SynthLoc` for evaluation, which contains four major groups of tasks `Open`, `Goto`, `PickUp`, and `PutNext`. The agent is asked to operate with an assigned object, like "open a red door" or "put the gray box in front of you next to the blue key". Tasks vary as the colors, types, or locations of objectives change, making around 6000 different tasks in total. To evaluate the algorithms' generalizations, we divide the task space into in-distribution tasks and out-of-distribution tasks. In-distribution tasks account for approximately 60% of the total number of tasks, with a total of 3325. For the offline learning, we construct an offline dataset with 50k expert trajectories, 50k imitation learning agent trajectories, and 25k random trajectories.

**ALFRED**  [54] benchmarks sequential decision-making tasks involving household activities (e.g, cleaning, heating food) through language instructions and first-person vision, shown on the Right of Figure 4. The dataset provides 8055 expert demonstrations with 25k human-annotated language instructions detailing both high-level goals and sub-goal step-by-step guidance. As our work primarily focuses on low-level policy learning rather than high-level planning, we specifically concentrate on the `GOTO` sub-goal setting for our evaluation. In this task set, the agent must go to specific locations according to instructions like "Move to other side of couch on the right side of the table before the door". To simulate the presence of noisy data in real-world applications, we augment the training set with 30k random-agent trajectories, resulting in 97896 total trajectories with 53442 unique instructions across 108 household scenes.

**Baselines**  We choose three state-of-the-art offline RL algorithms, CQL [30], IQL [28], and adapt them into a language-conditioned manner as our RL baselines. In addition, for imitation baselines, we include GCBC [15], BC-Z [25] and GRIF [42], which are also adapted into a language-conditioned manner by the benchmarks [10, 54]. Further details about baselines are shown in Appendix E.

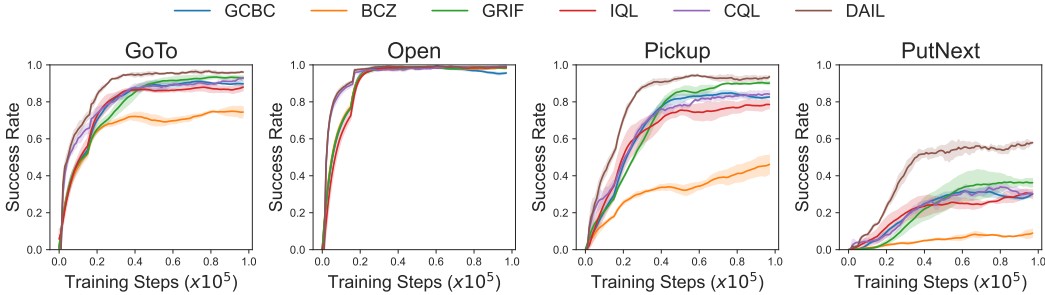

Figure 5: Training curves on in-distribution BabyAI tasks. Success rates are evaluated over 3 seeds.

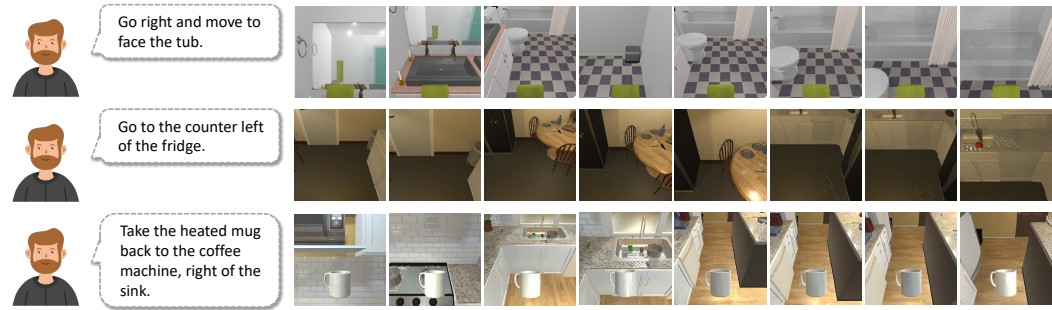

Figure 6: Example trajectories of DAIL in ALFRED validation set instructions with varied instructions and scenes.

## 5.2 Main Results

**BabyAI experimental results.** We provide training curves of in-distribution tasks in Figure 5 and out-of-distribution experimental results in Table 1. The complete results are shown in Table 5 in Appendix H. It shows that our method achieves superior performance compared with other baselines, especially in the `PutNext` task category. Vanilla offline RL algorithms like CQL and IQL underperform compared to imitation learning methods like GCBC, which we attribute to the adverse impact of task ambiguity on RL-based approaches as discussed in Theorem 1. On the other hand, modified algorithms designed for language-conditioned IL (BC-Z and GRIF) perform poorly under our setting. This is primarily because their contrastive learning objectives are not robust in the presence of noisy or suboptimal data. In contrast, our alignment-based approach, built on an offline RL framework, maintains strong performance. We further evaluate various algorithms on a more challenging dataset with fewer expert trajectories (Table 6 in Appendix H). Our method still achieves the optimal results. Further details about the experiment are shown in Appendix E.

**ALFRED experimental results.** The complexity and variety of instructions in ALFRED challenge the agent's ability to discern instructions. The experimental results in Table 2 show that our approach demonstrates the highest success rate (SR), validating its positive impact on instruction recognition capability compared to other baselines. GCBC exhibits poorer resistance to suboptimal data, resulting in performance comparable to other RL baseline models. We also report path-length weighted success rates (PLW SR), which considers the length of expert demonstration and demonstrates the effectiveness of the behavior policy following [54]. We further illustrate the observation trajectories generated by DAIL under the validation set instructions of ALFRED with varied instructions and scenes in Figure 6. This visualization demonstrates DAIL's robust task execution in complex scenes under diverse instructions, with additional trajectory demonstrations provided in Appendix I.

## 5.3 Visualization

To better understand how our proposed method enhances the performance, we show the t-SNE [57] results of the language instructions internal embedding on BabyAI `SynthLoc`. Each point repre-

Table 2: Success rate (SR) and path-length weighted success scores (PLW SR) in the ALFRED tasks. The results are shown on the training set and validation set respectively. Each score is evaluated over 3 seeds.

| Tasks | GCBC | BC-Z | GRIF | IQL | CQL | DAIL (ours) |
|---|---|---|---|---|---|---|
| SR (Training) | 87.9±2.4 | 86.5±0.8 | 87.8±1.6 | 88.4±0.6 | 87.2±1.2 | **92.3±2.2** |
| PLW SR (Training) | 84.3±2.6 | 82.3±2.7 | 82.3±2.6 | 84.4±2.7 | 83.0±2.9 | **90.2±3.1** |
| SR (Validation) | 47.1±2.4 | 43.0±2.5 | 48.6±1.0 | 52.0±3.0 | 50.4±1.7 | **56.8±2.1** |
| PLW SR (Validation) | 40.5±3.1 | 39.5±2.2 | 43.2±2.5 | 47.0±3.2 | 44.4±1.3 | **50.3±2.4** |

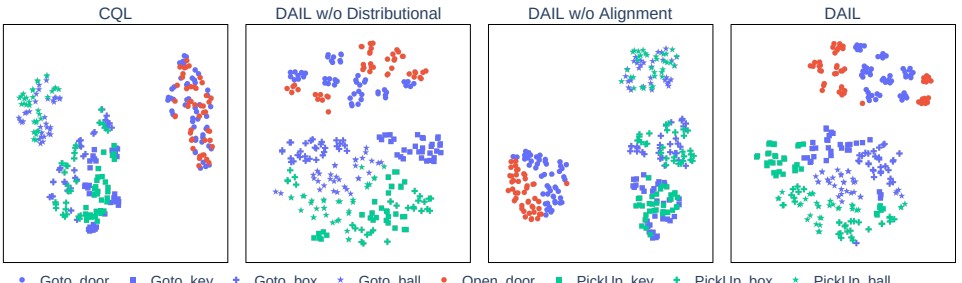

Figure 7: T-SNE visualization of instructions in BabyAI tasks. The figure distinguishes between different task categories (e.g., Open) and target object types (e.g., box), using marker colors and shapes to represent each separately. For example, "pick up a red box" corresponds to ■, "go to a green box" corresponds to +.

sents the internal representation of a unique language instruction within the policy network. The experimental results in Figure 7 show that vanilla RL can only marginally separate some broad task categories, but fails at distinguishing `PickUp` and `Goto`. Moreover, it is completely confused between `Open (door)` and `Goto (door)`. Alignment and distributional methods help discriminate tasks between and within categories. Our method substantially enhances task representation by clearly differentiating between task categories and target object types (for example, separating ● and ●, ⋆ and ⋆).

For more precise visualization, we use the same method to visualize the task representations of algorithms on the same task category, illustrated in Figure 17 in Appendix G. Our method effectively distinguishes all tasks in these two categories without overlapping confusion and group similar tasks into smaller clusters (■, ●, +, ⋆ and so on). We conduct the visualization experiments under 3 training seeds, all of which yield consistent experimental conclusions. Moreover, we quantitatively measure the clustering quality of our proposed components with the Silhouette score[52] using the learned language embeddings. The experimental results in Table 8 in Appendix H demonstrate that our method effectively enhances clustering performance, which aligns with the experimental findings from the visualization. Details of the quantitative evaluation and more visualizations of task representations can be referred in Appendix H and G, respectively.

## 5.4 Ablation Studies

**Ablation of components.** To study the contribution of each component in our learning framework, we conduct the following ablation study. We compare the performance of algorithms that only apply trajectory-wise alignment or distributional language-guided policy alone with our method on `SynthLoc`. The experimental results in Table 3 show that both modules significantly improve the performance over vanilla CQL on in-distribution and out-of-distribution tasks. Further, combining both components can achieve the best performance compared to other approaches.

**Ablation of alignment weight $\lambda$.** In Equation 13, $\lambda$ is the weight of the alignment loss. For this reason, we evaluate the choice of $\lambda$ in BabyAI tasks with various $\lambda$. The experimental results in Figure 8 show that there is no noticeable performance difference between $\lambda = 0.2$ and $\lambda = 1$. The

Table 3: Ablation results for components of our method. Each score is evaluated over 3 seeds.

| Algorithm | In Distribution | | Out of Distribution | |
|---|---|---|---|---|
| | PutNext | All | PutNext | All |
| CQL | 25.6±2.5 | 78.1±1.6 | 27.6±0.8 | 72.6±0.4 |
| DAIL w/o Distributional | 39.6±0.6 | 83.3±0.2 | 39.3±1.7 | 77.3±0.8 |
| DAIL w/o Alignment | 39.1±1.0 | 82.2±1.3 | 32.0±0.9 | 75.1±1.6 |
| DAIL | **57.9±0.9** | **89.2±0.5** | **49.1±1.8** | **81.7±1.3** |

performance begins to degrade when the influence of the loss is either too small ($\lambda = 0.01$) or too large ($\lambda = 2$). Therefore, we recommend choosing a value between 0.2 and 1.

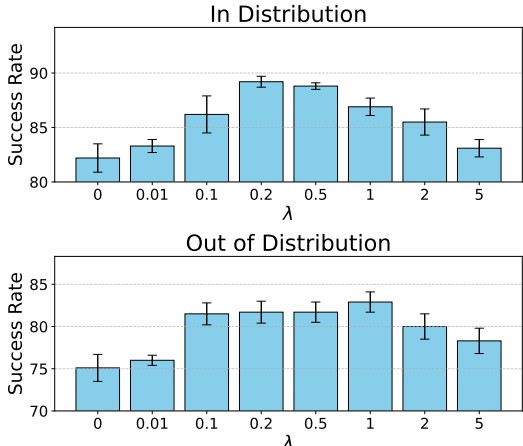

Figure 8: Percent difference of the performance of an ablation over $\lambda$, compared to the average results of all $\lambda$s evaluated.

## 6    Conclusion and Future Work

In this work, we aim to address the task discrimination and comprehension challenges in language-conditioned RL. To achieve this, we propose a novel method called DAIL, which incorporates a distributional language-guided policy and a trajectory-wise semantic alignment module. We first theoretically demonstrate that distributional RL methods are more sample-efficient than traditional RL methods for learning in language-conditioned tasks. We then perform extensive experiments on both structured and visual observation benchmarks. The experimental results and visualization analysis show that our method learns language instruction representations with clearer semantics. The simplicity and robustness of DAIL make it easily adaptable as a plug-in for other methods tackling language-conditioned problems. While our method is theoretically and empirically validated, demonstrating its significant advantages in language-conditioned tasks, several limitations remain. First, due to experimental constraints, we are unable to test our method in real-world scenes to further validate the method's effectiveness and robustness, which we will consider in future work. Second, our theoretical analysis of distributional RL's advantages relies on the assumptions of offline RL. Although we believe that this approach remains effective in online settings, we defer the theoretical analysis to future work.

## Acknowledgments and Disclosure of Funding

This work is supported by Strategic Priority Research Program of the Chinese Academy of Sciences (No.XDA27040200)

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

# A   Related Work

**Language-conditioned Agents.**   Two primary approaches for enabling an agent to follow human instructions are reinforcement learning (RL) and imitation learning (IL). Some approaches in language-conditioned RL focus on aligning language with policies or performing feature extraction to integrate language information into the learning process [9, 26, 1, 41, 53, 4], while others prioritize reward shaping [55, 16, 19, 18, 53] to formulate language-conditioned reward functions. However, most methods rely on simulators and are limited in scalability when applied in the offline setting. On the other hand, IL is designed to learn from large datasets but is heavily dependent on the quality of the data [24]. To mitigate this dependency and enhance data efficiency, recent works have leveraged crowd-sourced annotations [43], multimodal alignment [25, 42, 44], or carefully designed model architectures [39, 56]. These methods enhance IL by supplying richer supervisory signals or building more robust model structures. Despite the success of IL methods in solving complex tasks, the scale and quality of the dataset remain significant constraints, particularly without external annotations or auxiliary datasets.

**Offline Goal-conditioned RL.**   Learning a task-specific policy from demonstrations with different goals presents a significant challenge in offline goal-conditioned reinforcement learning (GCRL). By relabeling trajectories and treating intermediate states as additional goal states [2, 32, 33, 59, 60, 38], offline GCRL has achieved notable improvements in sample efficiency. However, this approach is challenging to replicate in language-conditioned settings. First, replicating a specific goal state is particularly challenging in environments characterized by randomness or partial observability. Second, directly using goal states that lack semantic context as labels proves ineffective for learning. While some works use language instructions to guide planning [13, 48, 46], the ambiguity of language can complicate this process. Several studies have applied hierarchical RL to guide policy through subgoals [32, 61, 8, 47, 34] and demonstrate strong performance. However, they rely heavily on a high-level policy that accurately decomposes language instructions into subgoals.

# B Algorithm

---

**Algorithm 1** Distributional Aligned Learning

---

**Require:** Offline dataset $\mathcal{D} = \{(\tau, l)\}$, target network update frequency $K_{\text{update}}$ and support atoms $Z_{\text{atoms}}$.

  Initialize policy parameters $\theta$ and target policy parameters $\hat{\theta}$.

  **for** each gradient step **do**

    Sample batch $\mathcal{B} = \{(\tau, l)\}_{i=1}^N \sim \mathcal{D}$

    Encode instructions: $\{x_l\}_{i=1}^N$

    Compute the history information $\{x_0, x_1, .., x_T\}_{i=1}^N$ using (8)

    // **Distributional Language-Guide Policy**

    **for** each transition $(s_t, a_t, r_t, s_{t+1}, l)$ in batch $\mathcal{B}$ **do**

      Compute estimated value distribution $Z_\theta(s_t, a_t, x_{t-1}, l)$ using (15)

      Compute projected update $\Phi\hat{\mathcal{T}}Z_\theta(s, a, x, l)$ using (18)

      $l_{\text{Dist}}(\theta) \leftarrow D_{KL}(\Phi\hat{\mathcal{T}}Z_{\hat{\theta}}(s_t, a_t, x_{t-1}, l)||Z_\theta(s_t, a_t, x_{t-1}, l))$

    **end for**

    // **Trajectory-Wise Semantic Alignment**

    **for** each trajectory-instruction pair $(\tau, l)$ **do**

      **for** each pair $(\tau', l')$ other than $(\tau, l)$ **do**

        View $l'$ as negative instruction $l^-$, and compute contrastive loss $l_c$ by using (9)

      **end for**

    **end for**

    Add up the losses above to get $\mathcal{L}_{\text{Dist}} \leftarrow \sum l_{\text{Dist}}$, and $\mathcal{L}_c \leftarrow \sum_{(\tau, l)} l_c$

    Compute conservative Q-learning loss $\mathcal{L}_{\text{CQL}}$ using (12)

    $\mathcal{L}_{\text{tot}} \leftarrow \mathcal{L}_{\text{Dist}} + \lambda\mathcal{L}_c + \alpha\mathcal{L}_{\text{CQL}}$

    Update $\theta \leftarrow \theta - \eta\nabla_\theta\mathcal{L}_{\text{tot}}(\theta)$

    **if** step % $K_{\text{update}}$=0 **then**

      Update target network: $\hat{\theta} \leftarrow \theta$

    **end if**

  **end for**

  Extract policy: $\pi(s, x, l) \leftarrow \arg\max_a \sum_{i=1}^M p_i(s, a, x, l)z_i$

  **return** $\pi(s, x, l)$

---

## C   Details on Distributional Language-Guided Policy

In this section, we provide a more detailed introduction to the specific computational workflow of our proposed Distributional Language-Guided Policy. We follow the approach in [3], employing discrete atoms to model and approximate the original value distribution. To facilitate understanding, we have included the calculation methodology for this section of the work here.

Specifically, we model the discrete value distribution with discrete units called atoms $\{z_i = V_{\text{MIN}} + i\Delta z\}_{i=0}^{M-1}$, which are uniformly spaced support points that discretize the range of possible returns into $[V_{MIN}, V_{MAX}]$. $M \in \mathbb{N}$ is the number of atoms. $V_{MAX}, V_{\text{MIN}}, M$ are pre-defined parameters, which together decide the step between the atoms of the categorical value distribution: $\Delta z := \frac{V_{\text{MAX}} - V_{\text{MIN}}}{M-1}$.

To represent the discrete value distribution, we need to estimate the probability $p_i$, which means the probability for the discrete value equaling $z_i$. In practical implementation, we approximate the probabilities $p_i$ by a parametric model $\theta : \mathcal{S} \times A \times \mathcal{X} \times \mathcal{L} \to \mathbb{R}^M$. We use $\theta_i(\cdot, \cdot, \cdot, \cdot)$ to denote the $i$-th dimension of this model's output. Therefore, the discrete value distribution can be written as:

$$
Z_\theta(s, a, x, l) = z_i \quad \text{w.p.} \quad p_i(s, a, x, l) := \frac{e^{\theta_i(s,a,x,l)}}{\sum_j e^{\theta_j(s,a,x,l)}}
$$

$$
Q_\theta(s, a, x, l) = \mathbb{E}[Z_\theta(s, a, x, l)] = \sum_{i=0}^{M-1} p_i(s, a, x, l) z_i \tag{15}
$$

where $\sum_{i=0}^{M-1} p_i = 1$. We now explain how to learn $\theta$ through RL. Given a transition $(s, a, r, s', l)$, the Bellman update for each atom $z_i$ is computed as:

$$
\hat{\mathcal{T}} z_i = r + \gamma z_i \tag{16}
$$

The Bellman update $\hat{\mathcal{T}} z_i$ maps the original support points to new locations that do not align with predefined discrete atoms $\{z_i\}_{i=0}^{M-1}$, making it impossible to represent as a valid discrete distribution over the fixed atoms. Therefore, we need to project these values back onto the fixed support $\{z_0, ..., z_{M-1}\}$. To do so, we distribute probability mass $p_i(s', a', x', l)$ to the nearest two atoms in $[V_{\text{MIN}}, V_{\text{MAX}}]$, where $a' = \pi(s', x', l)$ is the output of the greedy policy $\pi(\cdot, \cdot, \cdot)$. $x'$ is computed as $x' = h_w(x, (s, a))$. Ultimately, we can compute the projected probability for $\hat{\mathcal{T}} z_i$ via a local interpolation mechanism:

$$
\text{Projected probability} = \begin{cases} \frac{z_{j+1} - \hat{\mathcal{T}} z_i}{\Delta z} \cdot p_i(s', a', x', l), & \text{assigned to } z_j \\ \frac{\hat{\mathcal{T}} z_i - z_j}{\Delta z} \cdot p_i(s', a', x', l), & \text{assigned to } z_{j+1} \end{cases} \tag{17}
$$

Sum all the projected probabilities from all $\hat{\mathcal{T}} z_j$, we can compute the projected update probabilities by:

$$
(\Phi \hat{\mathcal{T}} Z_\theta(s, a, x, l))_i = \sum_{j=0}^{M-1} [1 - \frac{|[\hat{\mathcal{T}} z_j]_{V_{MIN}}^{V_{MAX}} - z_i|}{\Delta z}]_0^1 \, p_j(s', a', x', l) \tag{18}
$$

where $[\cdot]_a^b$ bounds the argument in the range $[a, b]$. Given the project update $\Phi \hat{\mathcal{T}} Z_{\hat{\theta}}$ and the current estimates of the discrete value distributions $Z_\theta$, we can update $\theta$ by minimizing the KL divergence:

$$
D_{KL}(\Phi \hat{\mathcal{T}} Z_{\hat{\theta}}(s, a, x, l) || Z_\theta(s, a, x, l)) \tag{19}
$$

where $\hat{\theta}$ is the target network.

# D  Theoretical Analysis

## D.1  Insights Using Distributional RL

The key insight of applying distributional RL to language-conditioned tasks lies in its capacity to capture value distributions, which provides fine-grained task differentiation across various tasks. Traditional RL methods, however, rely on learning scalar value expectations, discarding critical distributional information, making it require more samples to discriminate between tasks properly. We first demonstrate this key insight through an extreme yet illustrative example as illustrated in Figure 9.

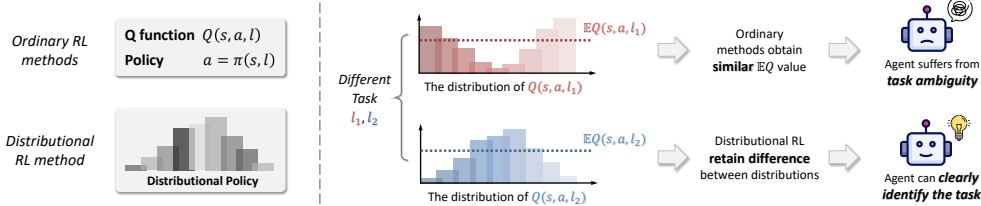

Figure 9: An illustrative case where $Q(s, a)$ functions in two tasks share the same expectations but have different distributions. In this case, traditional RL cannot discriminate between the tasks, while distributional RL can.

Consider two distinct tasks $l_1$ and $l_2$ that share identical expected returns $Q_\pi(s, a, l_1) = Q_\pi(s, a, l_2)$ for a specific state-action pair $(s, a)$, but exhibit fundamentally divergent Q value distributions $Z(s, a, l_1) \neq Z(s, a, l_2)$. In this case, traditional RL methods relying on value estimation would inevitably conflate tasks $l_1, l_2$ at $(s, a)$, regardless of sample size, while distributional RL resolves this ambiguity through approximating the full distributions of values.

To formally validate this insight, we present the following theorem, accompanied by a rigorous proof.

## D.2  Theoretical Proof

**Definition 1** (Restatement of Definition 1). *In a multi-task RL setting with known task instruction space $\mathcal{L}$ and unknown semantics space $\mathcal{G}$, for a task distinction threshold $\delta$ and sub-optimality gap $\epsilon$, two task instructions $l_i, l_j \in \mathcal{L}$ are considered with **different underlying semantics**, $g_i \nleftrightarrow g_j$, if the expected Q-values under any shared $\epsilon$-optimal policy $\pi$ satisfy:*

$$\mathbb{E}_\pi \left[ |Q_\pi(s, a, l_i) - Q_\pi(s, a, l_j)| \right] \geq \delta.$$

*Conversely, if*

$$\mathbb{E}_\pi \left[ |Q_\pi(s, a, l_i) - Q_\pi(s, a, l_j)| \right] \leq \delta/2,$$

*then $l_i$ and $l_j$ are considered with the **same underlying semantics**, $g_i \leftrightarrow g_j$, where $s_0 \sim p_0(\cdot), a \sim \pi(\cdot|s), s' \sim p(\cdot|s, a)$. $V_\pi(s) \geq V^*(s) - \epsilon, \forall s \in \mathcal{S}$, $V^*$ is optimal value function.*

*In the case of distributional reinforcement learning, the Wasserstein distance $W_1$ between the return distributions $Z(s, a)$ is used as the criterion. Specifically, $l_i$ and $l_j$ are considered to represent **different semantics** if*

$$\mathbb{E}_\pi \left[ W_1 \left( Z_\pi(s, a, l_i), Z_\pi(s, a, l_j) \right) \right] \geq d,$$

*and the **same semantics** if this expectation is less than or equal $d/2$.*

**Theorem 1** (Sample Complexity for Task Instruction Disambiguation). *Consider an offline multi-task RL setting with $M$ distinct tasks (with different semantics). In direct Q-value estimate setting, suppose $Q(s, a, l) \in [0, Q_{\max}], \forall(s, a, l) \sim \mathcal{D}$ with finite $Q_{\max}$, and the task distinction threshold is $\delta > 0$. When the number of training samples $n_{\text{value}}$ satisfies:*

$$n_{\text{value}} \geq \frac{C_{\text{value}} \log(3M^2/\eta)}{\delta^2}.$$

*The mean value estimate algorithm achieves task-level semantic disambiguation with confidence at least $1 - \eta$. In the distributional RL setting, let $Z(s, a, l)$ denote the learned return distribution, and*

*suppose the task distinction threshold is given by a 1-Wasserstein distance $d > 0$. Then, to ensure semantic disambiguation of task instructions with confidence at least $1 - \eta$, it suffices that:*

$$n_{\text{dist}} \geq \frac{C_{\text{dist}} \log(3M^2/\eta)}{d^2},$$

*where $C_{\text{value}}, C_{\text{dist}} > 0$ are universal constants depending on certain attributes of Q-value distribution.*

*Proof.* We denote $Q_i(s, a)$ as a shorthand for $Q_\pi(s, a, l_i)$ and $Z_i(s, a)$ as a shorthand for $Z_\pi(s, a, l_i)$ in the following discussion. For direct offline Q-learning, when the task distinction threshold is $\delta > 0$, we define the semantic ambiguity event $\mathcal{S}_a$ as:

$$\mathbb{E}_{(s,a) \sim \mathcal{D}} \left[ |\hat{Q}_i(s, a) - \hat{Q}_j(s, a)| \right] \leq \frac{\delta}{2}, \tag{20}$$

$$\mathbb{E}_\pi \left[ |Q_i(s, a) - Q_j(s, a)| \right] \geq \delta, g_i \nleftrightarrow g_j, \tag{21}$$

where $\mathcal{D}$ is the offline dataset, $Q$ is the optimal Q function, $\pi$ is the optimal policy along with $Q$, and $\hat{Q}$ is the learned Q function.

Following the triangle inequality, we have:

$$\left| \mathbb{E}_{(s,a) \sim \mathcal{D}}[|\hat{Q}_i - \hat{Q}_j|] - \mathbb{E}_\pi[|Q_i - Q_j|] \right| \leq \tag{22}$$

$$\left| \mathbb{E}_{(s,a) \sim \mathcal{D}}[|\hat{Q}_i - \hat{Q}_j|] - \mathbb{E}_\mathcal{D} \mathbb{E}_{(s,a) \sim \mathcal{D}}[|\hat{Q}_i - \hat{Q}_j|] \right| + \left| \mathbb{E}_\mathcal{D} \mathbb{E}_{(s,a) \sim \mathcal{D}}[|\hat{Q}_i - \hat{Q}_j|] - \mathbb{E}_\pi[|Q_i - Q_j|] \right| \tag{23}$$

In general, we assume that the offline RL dataset is collected by $\epsilon$-optimal policy, i.e., there exists a $\epsilon$-optimal policy $\pi$ that $\mathcal{D}$ satisfies

$$\mathbb{E}_\mathcal{D} \mathbb{E}_{(s,a) \sim \mathcal{D}}[|\hat{Q}_i - \hat{Q}_j|] = \mathbb{E}_\pi[|\hat{Q}_i - \hat{Q}_j|], \mathbb{E}_\mathcal{D}(\hat{Q}) = Q_\pi = Q \tag{24}$$

Besides, we have:

$$\left| \mathbb{E}_\pi[|\hat{Q}_i - \hat{Q}_j|] - \mathbb{E}_\pi[|Q_i - Q_j|] \right| = \left| \mathbb{E}_\pi \left( |\hat{Q}_i - \hat{Q}_j| - |Q_i - Q_j| \right) \right| \tag{25}$$

$$\leq \mathbb{E}_\pi \left| |\hat{Q}_i - \hat{Q}_j| - |Q_i - Q_j| \right| \tag{26}$$

$$\leq \mathbb{E}_\pi \left| (\hat{Q}_i - \hat{Q}_j) - (Q_i - Q_j) \right| \tag{27}$$

$$\leq \mathbb{E}_\pi |\hat{Q}_i - Q_i| + \mathbb{E}_\pi |\hat{Q}_j - Q_j| \tag{28}$$

Therefore,

$$\Pr \left( \left| \mathbb{E}_{(s,a) \sim \mathcal{D}}[|\hat{Q}_i - \hat{Q}_j|] - \mathbb{E}_\pi[|Q_i - Q_j|] \right| \geq \frac{\delta}{2} \right) \tag{29}$$

$$\leq \Pr \left( \left| \mathbb{E}_{(s,a) \sim \mathcal{D}}[|\hat{Q}_i - \hat{Q}_j|] - \mathbb{E}_\pi[|\hat{Q}_i - \hat{Q}_j|] \right| + \mathbb{E}_\pi |\hat{Q}_i - Q_i| + \mathbb{E}_\pi |\hat{Q}_j - Q_j| \geq \frac{\delta}{2} \right) \tag{30}$$

$$\leq \Pr \left( \left| \mathbb{E}_{(s,a) \sim \mathcal{D}}[|\hat{Q}_i - \hat{Q}_j|] - \mathbb{E}_\pi[|\hat{Q}_i - \hat{Q}_j|] \right| \geq \frac{\delta}{2} \right) \tag{31}$$

$$+ \Pr \left( \mathbb{E}_\pi |\hat{Q}_i - Q_i| \geq \frac{\delta}{2} \right) + \Pr \left( \mathbb{E}_\pi |\hat{Q}_j - Q_j| \geq \frac{\delta}{2} \right) \tag{32}$$

Since $\hat{Q}, Q \in [0, Q_{\max}]$, $|\hat{Q}_i - \hat{Q}_j|, |\hat{Q}_i - Q_i|, |\hat{Q}_j - Q_j| \in [0, Q_{\max}]$, together with Equation 24, we can apply Theorem 1 (Hoeffding's inequality) in [5], and obtain that for $\forall l_i, l_j \in \mathcal{L}$,

$$\Pr \left( \left| \mathbb{E}_{(s,a) \sim \mathcal{D}}[|\hat{Q}_i - \hat{Q}_j|] - \mathbb{E}_\pi[|\hat{Q}_i - \hat{Q}_j|] \right| \geq \delta/2 \right) \leq 2 \exp(-n_{\text{value}} \delta^2 / C_{\text{value}}) \tag{33}$$

$$\Pr \left( \mathbb{E}_\pi |\hat{Q}_i - Q_i| \geq \delta/2 \right) \leq 2 \exp(-n_{\text{value}} \delta^2 / C_{\text{value}}) \tag{34}$$

$$\Pr \left( \mathbb{E}_\pi |\hat{Q}_j - Q_j| \geq \delta/2 \right) \leq 2 \exp(-n_{\text{value}} \delta^2 / C_{\text{value}}) \tag{35}$$

where $n_{\text{value}} = |\mathcal{D}|$ is the size of dataset, $C_{\text{value}} \in (0, \mathcal{O}(Q_{\max}^2)]$ is a constant value.

Combining with Equation 32, we have

$$\Pr\left(\mathbb{E}_{(s,a)\sim\mathcal{D}}[|\hat{Q}_i - \hat{Q}_j|] - \mathbb{E}_\pi[|Q_i - Q_j|] \leq -\delta/2\right) \leq 6\exp(-n_{\text{value}}\delta^2/C_{\text{value}}) \tag{36}$$

Consider the extreme case in task semantics distinction, when $\exists l_i, l_j \in \mathcal{L}, g_i \nleftrightarrow g_j, \mathbb{E}_\pi[|Q_i - Q_j|] = \delta$ that can be exactly partitioned under the threshold. We have

$$\Pr\left(\mathbb{E}_{(s,a)\sim\mathcal{D}}[|\hat{Q}_i - \hat{Q}_j|] \leq \delta/2\right) \leq 6\exp(-n_{\text{value}}\delta^2/C_{\text{value}}) \tag{37}$$

Consider all task pairs with different real semantics $\begin{pmatrix} M \\ 2 \end{pmatrix} \approx \frac{M^2}{2}$:

$$\Pr\left(\exists i, j, g_i \nleftrightarrow g_j, \mathbb{E}_{(s,a)\sim\mathcal{D}}[|\hat{Q}_i - \hat{Q}_j|] \leq \delta/2\right) \leq 3M^2\exp(-n_{\text{value}}\delta^2/C_{\text{value}}) \tag{38}$$

To ensure a confidence level of at least $1 - \eta$ for any task ambiguity, we require that the probability of the event $\mathcal{S}_a$ satisfies:

$$\Pr(\mathcal{S}_a) \leq \eta,$$

Thus, we need to let:

$$3M^2\exp(-n_{\text{value}}\delta^2/C_{\text{value}}) \leq \eta.$$

Then,

$$n_{\text{value}} \geq \frac{C_{\text{value}}}{\delta^2}\log\frac{3M^2}{\eta} \tag{39}$$

For the distributional RL setting, we have the estimated return distributions $\hat{Z}_i, \hat{Z}_j$ and real distributions $Z_i, Z_j$, with the threshold $d > 0$. Similar to Equation 21, we have the semantic ambiguity event $\mathcal{S}_a$ as:

$$\mathbb{E}_{(s,a)\sim\mathcal{D}}\left[W_1(\hat{Z}_i(s,a), \hat{Z}_j(s,a))\right] \leq \frac{d}{2}, \tag{40}$$

$$\mathbb{E}_\pi\left[W_1\left(Z_i(s,a), Z_j(s,a)\right)\right] \geq d, g_i \nleftrightarrow g_j, \tag{41}$$

where $Z$ is the optimal distribution of Q-value, $\pi$ is the optimal policy along with $Z$, and $\hat{Z}$ is the learned distribution.

Similarly, we have:

$$\left|\mathbb{E}_{(s,a)\sim\mathcal{D}}\left[W_1(\hat{Z}_i, \hat{Z}_j)\right] - \mathbb{E}_\pi\left[W_1\left(Z_i, Z_j\right)\right]\right| \tag{42}$$

$$\leq \left|\mathbb{E}_{(s,a)\sim\mathcal{D}}\left[W_1(\hat{Z}_i, \hat{Z}_j)\right] - \mathbb{E}_\mathcal{D}\mathbb{E}_{(s,a)\sim\mathcal{D}}\left[W_1(\hat{Z}_i, \hat{Z}_j)\right]\right| \tag{43}$$

$$+ \left|\mathbb{E}_\mathcal{D}\mathbb{E}_{(s,a)\sim\mathcal{D}}\left[W_1(\hat{Z}_i, \hat{Z}_j)\right] - \mathbb{E}_\pi\left[W_1\left(Z_i, Z_j\right)\right]\right| \tag{44}$$

$$= \left|\mathbb{E}_{(s,a)\sim\mathcal{D}}\left[W_1(\hat{Z}_i, \hat{Z}_j)\right] - \mathbb{E}_\pi\left[W_1(\hat{Z}_i, \hat{Z}_j)\right]\right| + \left|\mathbb{E}_\pi\left[W_1(\hat{Z}_i, \hat{Z}_j) - W_1(Z_i, Z_j)\right]\right| \tag{45}$$

Following the triangle inequality of Wasserstein distance, we have:

$$W_1(\hat{Z}_i, \hat{Z}_j) - W_1(Z_i, Z_j) \leq W_1(\hat{Z}_i, Z_i) + W_1(Z_j, \hat{Z}_j) \tag{46}$$

Then,

$$\Pr\left(\left|\mathbb{E}_{(s,a)\sim\mathcal{D}}\left[W_1(\hat{Z}_i,\hat{Z}_j)\right]-\mathbb{E}_\pi\left[W_1(Z_i,Z_j)\right]\right|\geq\frac{d}{2}\right) \tag{47}$$

$$\leq\Pr\left(\left|\mathbb{E}_{(s,a)\sim\mathcal{D}}\left[W_1(\hat{Z}_i,\hat{Z}_j)\right]-\mathbb{E}_\pi\left[W_1(\hat{Z}_i,\hat{Z}_j)\right]\right|+\left|\mathbb{E}_\pi\left[W_1(\hat{Z}_i,\hat{Z}_j)-W_1(Z_i,Z_j)\right]\right|\geq\frac{d}{2}\right) \tag{48}$$

$$\leq\Pr\left(\left|\mathbb{E}_{(s,a)\sim\mathcal{D}}\left[W_1(\hat{Z}_i,\hat{Z}_j)\right]-\mathbb{E}_\pi\left[W_1(\hat{Z}_i,\hat{Z}_j)\right]\right|+\left|\mathbb{E}_\pi\left[W_1(\hat{Z}_i,Z_i)+W_1(\hat{Z}_j,Z_j)\right]\right|\geq\frac{d}{2}\right) \tag{49}$$

$$\leq\Pr\left(\left|\mathbb{E}_{(s,a)\sim\mathcal{D}}\left[W_1(\hat{Z}_i,\hat{Z}_j)\right]-\mathbb{E}_\pi\left[W_1(\hat{Z}_i,\hat{Z}_j)\right]\right|\geq\frac{d}{2}\right) \tag{50}$$

$$+\Pr\left(\left|\mathbb{E}_\pi\left[W_1(\hat{Z}_i,Z_i)\right]\right|\geq\frac{d}{2}\right)+\Pr\left(\left|\mathbb{E}_\pi\left[W_1(\hat{Z}_j,Z_j)\right]\right|\geq\frac{d}{2}\right) \tag{51}$$

Since $\hat{Z}$ is the empirical measure of $Z$ and the estimated return is 1-dimensional, $Q \in [0, Q_{\max}]$, $W_1(Z_i, Z_j) \in [0, Q_{\max}]$. Following Theorem 1 (Hoeffding's inequality) in [5], Corollary 5.2 and Remark 5 in [31], we have the following equation where $C_{\mathrm{dist}_1}, C_{\mathrm{dist}_2} \in (0, \mathcal{O}(Q_{\max}^2)]$ are constant values.

$$\Pr\left(\left|\mathbb{E}_{(s,a)\sim\mathcal{D}}\left[W_1(\hat{Z}_i,\hat{Z}_j)\right]-\mathbb{E}_\pi\left[W_1(\hat{Z}_i,\hat{Z}_j)\right]\right|\geq d/2\right)\leq 2\exp(-n_{\mathrm{dist}}d^2/C_{\mathrm{dist}_1}) \tag{52}$$

$$\Pr\left(\mathbb{E}_\pi\left[W_1(\hat{Z}_i,Z_i)\right]\geq d/2\right)\leq 2\exp(-n_{\mathrm{dist}}d^2/C_{\mathrm{dist}_2}) \tag{53}$$

$$\Pr\left(\mathbb{E}_\pi\left[W_1(\hat{Z}_j,Z_j)\right]\geq d/2\right)\leq 2\exp(-n_{\mathrm{dist}}d^2/C_{\mathrm{dist}_2}) \tag{54}$$

Combining with Equation 51, we can obtain the following equation where $C_{\mathrm{dist}} = \max(C_{\mathrm{dist}_1}, C_{\mathrm{dist}_2}) > 0$

$$\Pr\left(\mathbb{E}_{(s,a)\sim D}\left(W_1(\hat{Z}_i,\hat{Z}_j)\right)-\mathbb{E}_\pi\left(W_1(Z_i,Z_j)\right)\leq -d/2\right)\leq 6\exp(-n_{\mathrm{dist}}d^2/C_{\mathrm{dist}}) \tag{55}$$

In the same way, consider the extreme case when $\exists l_i, l_j \in \mathcal{L}, g_i \not\leftrightarrow g_j, \mathbb{E}_\pi(W_1(Z_i,Z_j)) = d$. We have

$$\Pr\left(\mathbb{E}_{(s,a)\sim D}\left(W_1(\hat{Z}_i,\hat{Z}_j)\right)\leq d/2\right)\leq 6\exp(-n_{\mathrm{dist}}d^2/C_{\mathrm{dist}}) \tag{56}$$

Consider all task pairs with different real semantics $\binom{M}{2}\approx\frac{M^2}{2}$:

$$\Pr\left(\exists i,j,g_i\not\leftrightarrow g_j,\mathbb{E}_{(s,a)\sim D}\left(W_1(\hat{Z}_i,\hat{Z}_j)\right)\leq d/2\right)\leq 3M^2\exp(-n_{\mathrm{dist}}d^2/C_{\mathrm{dist}}) \tag{57}$$

Similarly, to ensure a confidence level of at least $1-\eta$ for avoiding task ambiguity, we need

$$3M^2\exp(-n_{\mathrm{dist}}d^2/C_{\mathrm{dist}})\leq\eta.$$

Then,

$$n_{\mathrm{dist}}\geq\frac{C_{\mathrm{dist}}}{d^2}\log\frac{3M^2}{\eta} \tag{58}$$

$\square$

**Corollary 2.** *In a multi-task RL setting, to avoid task ambiguity with confidence level $1-\eta$, learning the distribution over Q-values requires fewer samples than learning point estimates of Q-values when the number of tasks $M$ is sufficiently large. Formally, $n_{\mathrm{value}}\geq n_{\mathrm{dist}}$, where $n_{\mathrm{value}}, n_{\mathrm{dist}}$ denote the samples needed to avoid task ambiguity for value-based and distributional settings.*

*Proof.* From Theorem 1, we have

$$n_{\mathrm{value}}\geq\frac{C_{\mathrm{value}}}{\delta^2}\log\frac{3M^2}{\eta} \tag{59}$$

$$n_{\mathrm{dist}}\geq\frac{C_{\mathrm{dist}}}{d^2}\log\frac{3M^2}{\eta} \tag{60}$$

First, $Q \in [0, Q_{\max}], W_1(Z_i, Z_j) \in [0, Q_{max}]$, following Theorem 1 (Hoeffding's inequality) in [5] , Corollary 5.2 and Remark 5 in [31] we obtain that the constant $C$ satisfies:

$$C_{\text{value}} \leq \mathcal{O}(Q_{\max}^2), C_{\text{dist}} \leq \mathcal{O}(Q_{\max}^2). \tag{61}$$

Then, we can prove that for any Q-value distribution $Z_i, Z_j$,

$$W_1(Z_i, Z_j) \geq |\mathbb{E}_{Z_i}(Q) - \mathbb{E}_{Z_j}(Q)| \tag{62}$$

The definition of Wasserstein-1 distance is

$$W_1(Z_i, Z_j) = \inf_{\pi \in \Pi(Z_i, Z_j)} \mathbb{E}_{(X,Y) \sim \pi} |X - Y|$$

Here, $\Pi(Z_i, Z_j)$ denotes the set of all joint distributions with marginals $Z_i, Z_j$.

For $(Z_i, Z_j) \sim \pi$, we have

$$\mathbb{E}[|X - Y|] \geq |\mathbb{E}(X - Y)| = |\mathbb{E}_{Z_i}(X) - \mathbb{E}_{Z_j}(Y)| = |\mathbb{E}_{Z_i}(X) - \mathbb{E}_{Z_j}(X)|$$

Thus, for any joint $\pi$,

$$\mathbb{E}_{(X,Y) \sim \pi} |X - Y| \geq |\mathbb{E}_{Z_i}(X) - \mathbb{E}_{Z_j}(X)|$$

For the optimal joint $\pi$,

$$W_1(Z_i, Z_j) \geq |\mathbb{E}_{Z_i}(Q) - \mathbb{E}_{Z_j}(Q)| \tag{63}$$

Thus, we must choose a much smaller threshold $\delta$ for point estimates than that $d$ for distribution learning, $\delta \leq d$. In some scenarios, when the mean difference of the Q function is small but the distribution difference is large, $\delta << d$.

Combining Equation 61, 63, we can obtain that

$$\frac{C_{\text{value}}}{\delta^2} \log \frac{3M^2}{\eta} \geq \frac{C_{\text{dist}}}{d^2} \log \frac{3M^2}{\eta} \tag{64}$$

we prove that $n_{\text{value}} \geq n_{\text{dist}}$, even $n_{\text{value}} >> n_{\text{dist}}$ in some scenarios.

$\square$

# E  Experimental Details

## E.1  Toy Experiment

In this toy experiment based on the Minigrid environment [11], we demonstrate that vanilla offline RL fails to establish the relationship between the tasks and their underlying reward functions as the number of tasks explodes.

The setup consists of 10 accessible goal positions $\mathcal{G} = \{g_0, g_1, ..., g_9\}$, where $\mathcal{G}$ represents the set of all possible goal positions. The agent (red triangle-shaped) must follow a given instruction $l \in \mathcal{L}$ to navigate to a specific goal position. We simulate instructions using numerical task IDs, making $\mathcal{L} \subset \mathbb{N}$. We employ a random mapping $F : \mathcal{L} \to \mathcal{G}$ to assign each $l$ to goal position $g$, which simulates semantics of instructions and is hidden from the agent. With these settings, we can control the number of instructions $|\mathcal{L}|$ by simply adjusting the set of valid task IDs and the mappings. We conduct 10 experiments, varying the number of instructions from 1 to $2^9$. For each experiment, we generate a new mapping $F$ and collect 1024 random trajectories as the offline dataset.

The agent always starts from the center of the map. For each step, the agent receives the task ID and current state as input, and the agent can choose to move forward, turn left, or turn right. A reward of $1 - 0.9 \times (\texttt{STEP\_COUNT}/\texttt{MAX\_STEP})$ is given for success and 0 for failure. $\texttt{MAX\_STEP}$ is fixed to 12 in our toy experiment. For offline datasets, a random policy with $\texttt{MAX\_STEP} = 12$ is used to generate

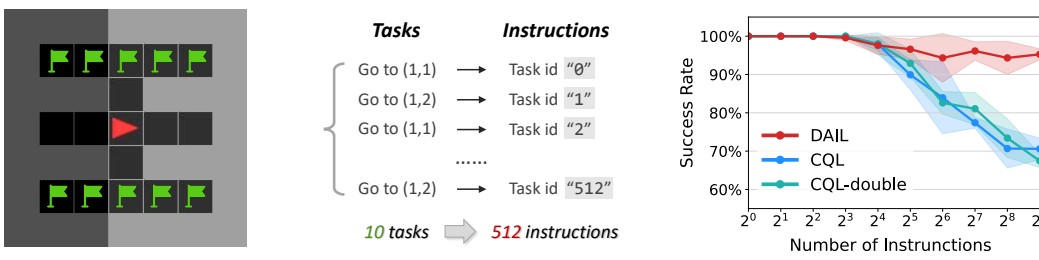

Figure 10: **Left:** The green flags in the map denote the accessible goals. **Middle:** An illustration of the mapping between goal positions and instructions. **Right:** Average success rates over 100 evaluations for each number of instructions and 3 seeds.

$64 \times n$ trajectories for 8 or fewer tasks, where $n$ is the number of tasks, and 1024 trajectories for 16 or more tasks, respectively. For all these datasets, the success rate is fixed to be 0.5.

For the observation signals in this toy experiment, we use a simple architecture, which combines Bag-of-Words encoding [40] and a two-layer CNN. We use 1 fully connected layer to embed task IDs into task representations and a two-layer MLP as the output network. We use 64 as the feature size for CQL and our method, 128 for CQL-double separately. For each number of tasks, each agent is trained with $\alpha = \{0, 0.01, 0.1, 0.5, 1, 2\}$ over 3 seeds. We calculate the average of the results from these 3 seeds and record the best performance among them as the performance for this number of instructions, as illustrated in the Right of Figure 10.

## E.2  Offline Datasets

### E.2.1  BabyAI

BabyAI [10] is a language-conditioned research platform built on MiniGrid [11], which provides different levels of tasks equipped with varied language instructions. We choose level `SynthLoc` as the benchmark, which is the union of all instructions from `PutNext`, `Open`, `Goto`, and `PickUp`. The agent needs to deal with synthetic Baby Language and interact with the specified objects at the goal position. Some examples of language instructions are "put the green key behind you next to a box", "go to the red ball behind you", and "pick up a green box".

We follow the default map configuration of BabyAI, where each room has a size of $7 \times 7$, arranged in a $3 \times 3$ grid with a total of 9 rooms. Each room may be connected to others via a door, as illustrated in Figure 11. The agent has a field of view of $7 \times 7$ in front of it, with the observation size being $(7 \times 7 \times 3)$. Each grid cell in the observation contains the values (`OBJECT_IDX`, `COLOR_IDX`, `STATE`),



Figure 11: Level `SynthLoc` in BabyAI

all represented in a structured format. A reward of '1 - 0.9 * (step_count / max_steps)' is only given for success, and '0' in all other cases. Agents are permitted to take up to 300 steps before truncation in our setting. In `SynthLoc`, the total number of tasks is large, and their distribution is sparse. We

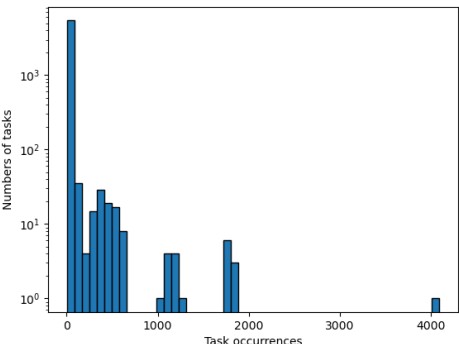

Figure 12: The histogram shows the frequency distribution of tasks in $10^6$ samples. The x-axis represents the total frequency of each task, while the y-axis indicates the number of tasks that fall within each frequency interval.

reset the environment $10^6$ times and obtain over 5500 unique instructions. From the histogram of task count (Figure 12), it can be observed that most tasks only appear less than 500 times in $10^6$, while some tasks appear over 4000 times. The distribution of tasks highlights the highly uneven distribution of tasks. We divide the task set into two subsets, designating approximately 60% of the tasks as in-distribution tasks. All trajectories in the offline dataset are collected under in-distribution instructions, while tasks encountered during testing outside this set are considered out-of-distribution tasks.

To construct the offline dataset, we collect three types of data: expert data, gathered by a pre-designed bot within the environment; medium data, collected by a well-trained agent; and random data. The built-in bot has access to global information to accomplish every possible task with a near-optimal solution. We train an IL agent following BabyAI 1.1 [23], the state-of-the-art model proposed by the original environmental authors. Trained on a dataset of 100k expert trajectories, it achieved approximately 87.9% success rate across all tasks. Random agent achieves a 10.5% success rate during data collection. We conduct a high-quality dataset with 50k expert trajectories, 50k IL agent trajectories, and 25k random trajectories; a medium-quality dataset with 12.5k expert trajectories, 25k IL agent trajectories, and 40k random trajectories. All the trajectories in the dataset are generated under in-distribution instructions.

### E.2.2 ALFRED

ALFRED [54] benchmarks sequential decision-making tasks involving household activities (e.g., cleaning, heating food) through language instructions and first-person vision, shown in Figure 13 and Table 4. The dataset provides 8055 expert demonstrations with 25k human-annotated language instructions detailing both high-level goals and sub-goal step-by-step guidance. As our work primarily focuses on low-level policy learning rather than high-level planning, we specifically concentrate on the GOTO sub-goal setting for our evaluation. In this task set, the agent must go to specific locations according to instructions like "Move to other side of couch on the right side of the table before the door". To simulate the presence of noisy data in real-world applications, we augment the training set with 30k random-agent trajectories, resulting in 97896 total trajectories with 53442 unique instructions across 108 household scenes.

As for the experiment, we use the Modeling Quickstart dataset, which is recommended [54], including trajectory JSONs and pre-generated ResNet features. The ResNet features are obtained using a pre-trained ResNet-18 [21] to extract $512 \times 7 \times 7$ features from the conv5 layer, which are used as observation input during training and evaluation.

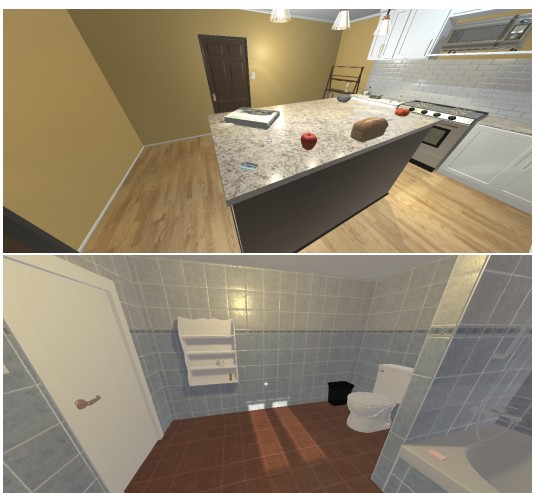

Figure 13: Two example scenes from ALFRED.

Table 4: Instruction examples in ALFRED GOTO task set.

| # | Instructions |
|---|---|
| 1 | Go left and turn to the right to face the couch. |
| 2 | Turn around and make a left immediately after the toilet turn a quick left to face the side of the toilet. |
| 3 | Move to other side of couch on the right side of the table before the door. |
| 4 | go back to your right to the fridge and open the door |
| 5 | Turn around and walk to the white stove on the right. |
| 6 | Turn to the right and go to the sink across from you. |
| 7 | Turn around and walk towards the toilet, then turn right and walk towards the door, turn left to face the counter. |
| 8 | Walk forward, then hang a right and walk across the room, turn left and walk up to the chair. |

For sub-goal evaluation, we follow [54] to use the expert trajectory to move the agent until the start state of the tested sub-goal, and the agent takes over to operate based on the instructions and observations. Episodes with 5 or more failed actions or exceeding the MAX_STEP = 32 are counted as failures immediately. A reward of 1 is given for success and 0 for failure. Failed actions refer to actions that cannot be successfully executed in the current state (for example, moving against the walls or other obstacles in the room).

# F  Architecture, Training, and Evaluation Details

## F.1  Details of Encoders

This section describes the network architecture of we used for language and observation encoding in BabyAI and ALFRED environments. In all the experiments, all models share the same encoders, detailed as follows, unless otherwise specified.

**Language Encoder**  For the language signals, we use the Transformer model [58] from the pre-trained CLIP network [50] for language encoding in BabyAI and ALFRED experiments, freezing the entire model during training and adding an additional fully connected layer at the end for fine-tuning.

**Observation Encoder**  Given the differing input structures of BabyAI and ALFRED, we employ separate observation encoders.

For **BabyAI**, we adopt the original visual encoding framework from BabyAI [23], which integrates a Bag-of-Words embedding layer [40], a convolution backbone, and a linear layer. The BOW module first turns the structural inputs with size $7 \times 7 \times 3$ into $7 \times 7 \times 256$ embeddings. The subsequent convolutional backbone processes these features through two sequential blocks and a max-pooling layer: each block contains a $3 \times 3$ convolutional layer, followed by batch normalization and ReLU activation. The output is then processed with a $7 \times 7$ max-pooling layer. The features are then flattened and projected to 256 dimensions through a linear layer, producing a 256-dimensional vector as the observation encoder's final output.

For **ALFRED**, we use the original encoding framework in ALFRED [54] for all implemented methods, which contains two sequential blocks: each block contains a $1 \times 1$ convolutional layer, followed by batch normalization and ReLU activation. The features are then flattened and projected to 512 dimensions through a linear layer, producing a 512-dimensional vector as the observation encoder's final output.

**FiLM and Sequence Encoder**  We follow [23] to use FiLM [49] to fuse language and observation encodings through feature-wise affine transformations. For history encoding, all baseline methods employ a two-layer unidirectional LSTM to model temporal dependencies.

## F.2  Architecture Details of DAIL

The overview architecture of DAIL is shown in Figure 14. We adopt the language and observation encoders described above to encode instructions and observations separately. To use the same sequence encoder for both trajectory-wise semantic alignment and history encoding, we modify the sequence encoder to process observation-action trajectories jointly and output history information $x_t$.

The computation process of state-action value $q(s_t, a_t, x_{t-1}, l)$ is shown on the Left of Figure 14, given instruction $l$, observation $s_t$. The outputs of the FiLM network are concatenated with the outputs of the sequence encoder, then processed through a Multi-Layer Perceptron (MLP) and a Softmax layer to generate the final value distribution with dimension $M$. For Trajectory-Wise Semantic Alignment as shown on the Right of Figure 14, we derive embeddings from instruction $l$ and trajectory $\tau$. The trajectory embedding is represented by the sequence model's final output $x_\tau$.

## F.3  Training Details

All models are implemented with PyTorch, and trained with a batch size of 64, using the Adam optimizer [27] at a learning rate of $3e - 4$. All layers in the networks utilize PyTorch's default weight initialization, and the network outputs fixed-dimensional embeddings suitable for downstream tasks. In BabyAI experiments, all methods were trained for 50 epochs over 3 seeds. And in the ALFRED experiments, all methods were trained for 20 epochs over 3 seeds following [54].

As for DAIL, we fix $\alpha = 2$ and $\lambda = 0.2$ except for the toy experiment and ablation experiment of $\lambda$. We use $V_{MAX} = -V_{MIN} = 20$, $M = 51$ in all our experiments following [29].

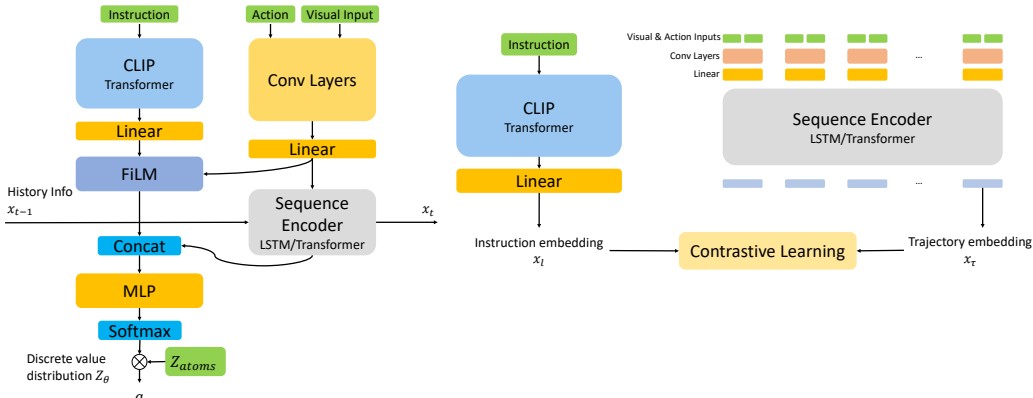

Figure 14: Overview of our algorithm. **Left:** Computation process of state-action value. **Right:** trajectory-wise semantic alignment.

### F.4 Baseline Details

In this work, all the baselines share the same language encoder and similar observation encoder, and only differ in the decision module. In the following part, we introduce the decision modules of several baselines used in our paper:

**GCBC:** language-conditioned behavior cloning with all data to maximize

$$\mathcal{J}_{\mathrm{BC}}(\pi_\theta) = \mathbb{E}_{(s_t,a_t,l)\sim\mathcal{D},x_t\sim h_w}[\log \pi_\theta(a_t|x_t,l)] \tag{65}$$

**BC-Z:** learning two task encodings from language and trajectory (video), and aligning them through similarity. We use cosine distance $D_{cos}(p,q) = \frac{p^T q}{|p||q|}$ to measure the similarity. We denote $(\tau, l)$ as a pair of trajectory and instruction.

$$\mathcal{J}_{\mathrm{BC-Z}}(\pi_\theta) = \mathbb{E}_{(s_t,a_t,l)\sim\mathcal{D},x_t\sim h_w}[\log \pi_\theta(a_t|x_t,f_\varphi(l))] - \mathbb{E}_{(\tau,l)\sim\mathcal{D}}[D_{\cos}(q_\phi(\tau),f_\varphi(l))], \tag{66}$$

where $q_\phi(\cdot)$ is the video encoder proposed by BC-Z to encode trajectory information, and $f_\varphi(\cdot)$ is the instruction encoder.

**GRIF:** explicitly aligning the representations of language-conditioned tasks through contrastive learning with similarity measure $\mathcal{C}(s,g,l) = D_{cos}(f_\varphi(l), h_\psi(s_0,g))$, where $\varphi,\psi$ are learnable parameters of the language encoder and goal encoder respectively and $g$ is the last state of a trajectory. Positive data $(\tau^+, l^+) \sim p_\mathcal{D}(\cdot,\cdot)$ are uniformly sampled from the dataset, with $s^+, g^+$ being the start state and end state of $\tau^+$ respectively. Negative examples $s^-, g^-$ are the start state and end state of a randomly sampled trajectory from the dataset. Negative instruction $l^- \sim p_l(\cdot)$ is the instruction of another random trajectory. For each positive example, $k$ negative examples are sampled noted as $\{s_i^-, g_i^-\}_{i=1}^k$ and $\{l_i^-\}_{i=1}^k$.

$$\mathcal{L}_{\mathrm{lang}\to\mathrm{goal}}(\varphi,\psi) = -\log \frac{\exp(\mathcal{C}(s^+,g^+,l^+)/\tau)}{\exp(\mathcal{C}(c^+,g^+,l^+)/\tau) + \sum_{i=1}^k \exp(\mathcal{C}(c_i^-,g_i^-,l^+)/\tau)}$$
$$\mathcal{L}_{\mathrm{goal}\to\mathrm{lang}}(\varphi,\psi) = -\log \frac{\exp(\mathcal{C}(s^+,g^+,l^+)/\tau)}{\exp(\mathcal{C}(c^+,g^+,l^+)/\tau) + \sum_{i=1}^k \exp(\mathcal{C}(c_i^+,g_i^+,l^-)/\tau)} \tag{67}$$

where $\tau$ is the temperature parameter. We employ the last state $s_T$ in the trajectory as the goal state: $h_\psi(s_0,g) = h_\psi(s_0,s_T)$. Then the policy network is trained with behavior cloning by maximizing the likelihood of the actions:

$$\mathcal{J}_{\mathrm{GRIF}}(\pi_\theta) = \mathbb{E}_{(s_t,a_t,l)\sim\mathcal{D},x_t\sim h_w}[\log \pi_\theta(a_t|x_t,f_\varphi(l))] \tag{68}$$

We use the **GRIF(Joint)** setting to train the model [42].

**CQL:** we implement CQL (w/o distributional) based on DDPG,

$$\mathcal{J}_{CQL}(\pi_\theta) = \mathbb{E}_{(s_t,a_t,l)\sim\mathcal{D},x_t\sim h_w}[Q(x_t,\pi_\theta(x_t,l),l)] \tag{69}$$

where Q function is learned by minimizing:

$$
\begin{aligned}
\mathcal{L}_{CQL(\mathcal{H})}(\theta) =& \mathbb{E}_{(s_t,a_t,r_t,s_{t+1},l)\sim\mathcal{D},(x_t,x_{t+1})\sim h_w}[(Q_\theta(x_t,a_t,l) - \mathcal{B}^\pi Q_\theta(x_t,a_t,l))^2] + \\
& \alpha\mathbb{E}_{(s_t,l)\sim\mathcal{D},x_t\sim h_w}[\log\sum_a \exp(Q_\theta(x_t,a,l)) - \mathbb{E}_{a\sim\hat{\pi}_\beta(a|s_t,l)}[Q_\theta(x_t,a,l)]]
\end{aligned}
\tag{70}
$$

where $\hat{\pi}_\beta$ is the behavior policy, $\mathcal{B}^\pi$ is the Bellman operator, and the balance ratio $\alpha = 2$ in our experiment.

**IQL:** training an additional value network and extracting policy through advantage weighted regression.

$$
\begin{aligned}
\mathcal{L}_V(\psi) &= \mathbb{E}_{(s_t,a_t,l)\sim\mathcal{D},x_t\sim h_w}[L_2^\tau(Q_{\hat{\theta}}(x_t,a_t,l) - V_\psi(x_t,l)] \\
\mathcal{L}_Q(\phi) &= \mathbb{E}_{(s_t,a_t,r_t,s_{t+1},l)\sim\mathcal{D},(x_t,x_{t+1})\sim h_w}[(r_t + \gamma V_\psi(x_{t+1},l) - Q_\phi(x_t,a_t,l)^2] \\
\mathcal{J}(\pi_\theta) &= \mathbb{E}_{(s_t,a_t,l)\sim\mathcal{D},x_t\sim h_w}[\exp(\beta Q_\phi(x_t,a_t,l) - V_\psi(x_t,l))\log\pi_\theta(a|x_t,l)]
\end{aligned}
\tag{71}
$$

The expectile $\tau = 0.7$, $\beta = 5$. We follow the authors' suggestions and subtract 1 from the reward if it equals 0.

To follow the original IL setting and simplicity, we merely use observation without action information in the history state encoding in BC and IQL. Our primary experiments show that it has little impact on the results. We also investigate recent approaches of language-conditioned IL such as LLfP [37] and R3M [44], but they perform poorly in prior experiments, so only BC-Z is chosen as the representative in the final baselines.

# G Visualization Supplementary Results

We apply t-SNE visualization in different algorithms and task types to show that our method substantially improves task representation on offline language-conditioned RL. Beyond the main text, here are some supplementary visualization results.

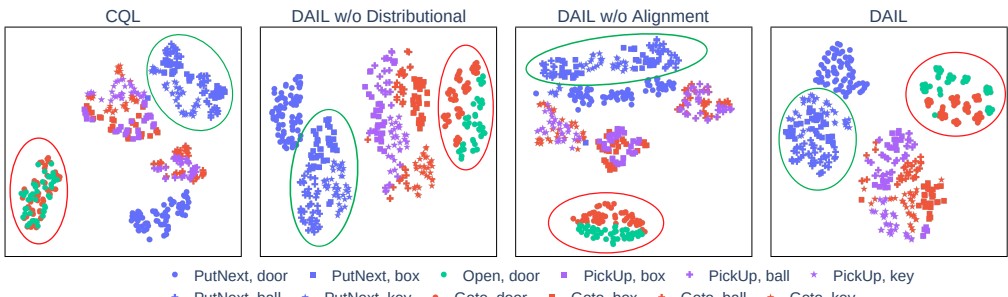

Figure 15: The t-SNE visualization of instructions from various tasks in BabyAI for different algorithms. The figure distinguishes between different **task categories** (e.g., PutNext) and **target object types** (e.g., box), using marker colors and shapes to represent each separately.

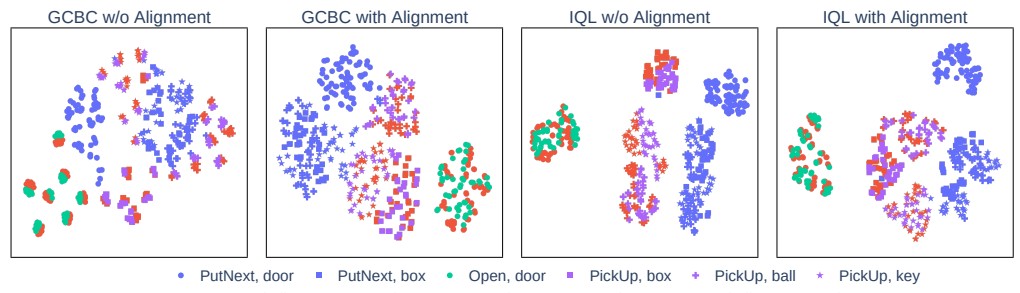

Figure 16: The t-SNE visualization of instructions from various tasks in BabyAI for different algorithms. The figure distinguishes between different **task categories** (e.g., PutNext) and **target object types** (e.g., box), using marker colors and shapes to represent each separately.

**Overall task representation** As introduced in Section 5, in BabyAI `SynthLoc` tasks, there are four main categories of tasks: `Goto`, `PickUp`, `PutNext`, and `Open`. We sample tasks from all four categories to visualize in Figure 15. Our method demonstrates superior task embedding capabilities compared to other approaches in the `Goto`, `PickUp`, and `Open`—effectively reducing confusion as highlighted by the red circles. Some degree of confusion remains inevitable in the `PutNext` tasks, due to their complexity and variability (as indicated by the green circles).

We also visualize representations from GCBC and IQL (with and w/o alignment) in Figure 16. GCBC shows reliable task representation, but confuses tasks from `Goto` and `PickUp`, "Open, door" ● and "Goto, door" ●, where their embeddings are tightly gathered into small clusters. Our further results in Figure 18 show that each cluster represents a specific color. Similarly, alignment significantly eases this issue by separating each instruction. IQL shows similar results to CQL, while alignment has a more significant influence on CQL. This result explains why simple GCBC shows comparable performance in our settings while vanilla offline RL fails due to confusion in task encoding.

**Task `PickUp` and `Goto`** We take a closer look at tasks in `GoTo` and `PickUp` that only have one target object. The task representations of DAIL and ablation algorithms are shown in Figure 17, and GCBC and IQL (with and w/o alignment) are shown in Figure 18. The color of the markers indicates that of the target objects other than black markers. As illustrated in Figure 7, our method further subdivides tasks (for example, ● and ●) into smaller clusters. Upon closer observation, these smaller clusters correspond to different target object colors. Similarly, CQL performs poorly in object color recognition, while distributional representation and semantic alignment substantially help task discrimination. As illustrated by the red circles, when the target object type is key, only our method

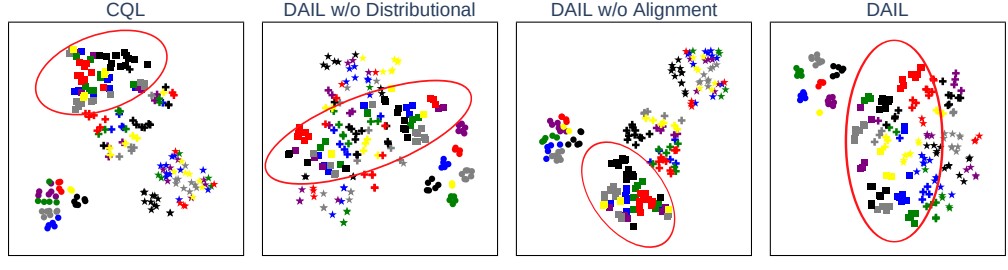

Figure 17: The t-SNE visualization of instructions from the same task categories with more detailed distinction. The figure distinguishes between different **target object types** (e.g., door) and **target object colors** (e.g., blue), using marker colors and shapes to represent each separately. For example, "go to the red door" corresponds to ●;. "go to a red ball behind you" corresponds to ⋆, "pick up the ball in front of you" corresponds to ⋆.

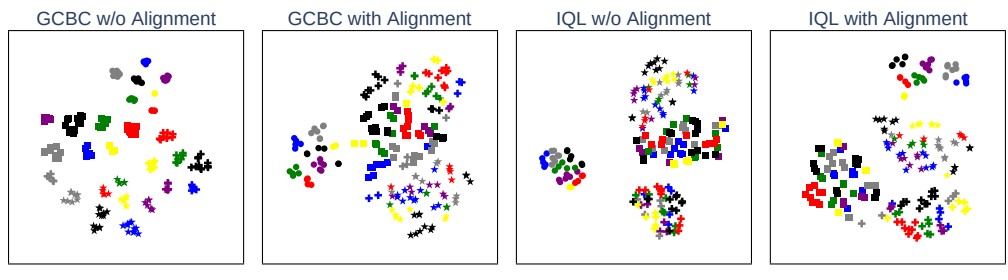

Figure 18: The t-SNE visualization of instructions from the same task categories (`PickUp` and `Goto`) with more detailed distinction. The figure distinguishes between different **target object types** (e.g., door) and **target object colors** (e.g., blue), using marker colors and shapes to represent each separately. The legend has the same meaning as in Figure 17.

succeeds in separating embeddings of different target colors while forming clusters for targets of the same color. In contrast, other methods tend to produce entangled representations, leading to less distinguishable task embeddings.

GCBC yields strong results in this scenario, whereas IQL relatively performs poorly. However, excessive overlap in GCBC suggests a confusion between the `PickUp` and `Goto` tasks. IQL can distinguish between different types of targets (different shapes of markers) but fails in differentiating target colors (for example, ■ ■ ■ . . . ). After our alignment method was applied, its performance significantly improved.

**Representation with instruction texts** We add some instruction texts to the representation map to better demonstrate the language instructions' representation results. We draw around 300 instructions from BabyAI `SynthLoc` level to visualize and sample around 20 tasks and include their original text in the figure, displayed to the right of the corresponding marker. Figure 19 shows the detailed representation result of vanilla CQL, and Figure 20 shows that of our method DAIL. The red dashed circles highlight the "Goto, door ●" and "Open, door +" tasks. Our method DAIL successfully separates the two task categories and further organizes them into clusters (based on colors, see Figure 17) while CQL fails. Similarly, the green dashed circles denote the "PickUp, key ●" and "Goto, key ■" tasks. While CQL fails to disentangle the `PickUp` and `Goto` task types, our method achieves a clear separation between them.

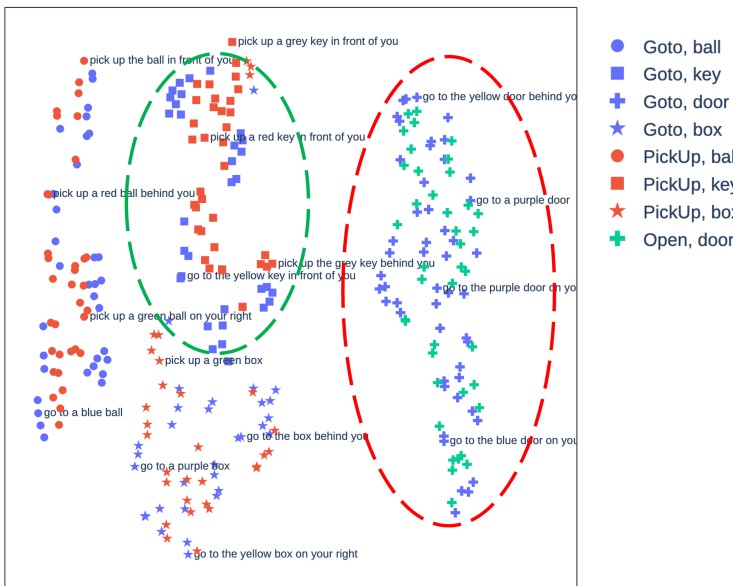

Figure 19: The t-SNE visualization of instructions with text annotations from `Open`, `Goto`, `PickUp` in BabyAI for CQL. The figure distinguishes between different **task categories** (e.g., PickUp) and **target object types** (e.g., box), using marker colors and shapes to represent each separately.

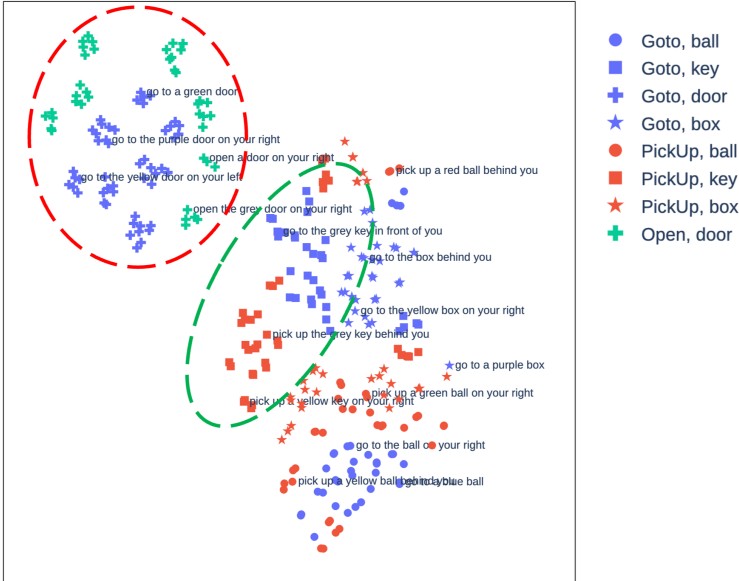

Figure 20: The t-SNE visualization of instructions with text annotations from `Open`, `Goto`, `PickUp` in BabyAI for our method. The figure distinguishes between different **task categories** (e.g., PickUp) and **target object types** (e.g., box), using marker colors and shapes to represent each separately.

# H   Additional Results

**Results on the hight-quality dataset of BabyAI**   The success rates of in-distribution and out-of-distribution tasks on the high-quality dataset are shown in Table 5. in both in-distribution and out-of-distribution tasks. Our method demonstrates significant advantages over other approaches on out-of-distribution tasks, particularly achieving substantial performance improvements on complex tasks such as `PutNext` compared to the baseline RL method CQL (49.1% vs. 27.6%). Vanilla offline RL algorithms like CQL and IQL underperform compared to imitation learning methods like GCBC, which we attribute to the adverse impact of task ambiguity on RL-based approaches as discussed in Theorem 1. On the other hand, modified algorithms designed for language-conditioned IL (BC-Z and GRIF) perform poorly under our setting. This is primarily because their contrastive learning objectives are not robust in the presence of noisy or suboptimal data. In contrast, our alignment-based approach, built on an offline RL framework, maintains strong performance.

Table 5: Success rate of in-distribution tasks and out-of-distribution BabyAI tasks. Each score is evaluated over 3 seeds.

| Algorithm | Open | Goto | PickUp | PutNext | All |
|---|---|---|---|---|---|
| *In Distribution* | | | | | |
| GCBC | 96.9±0.8 | 91.8±1.1 | 85.6±0.0 | 27.6±3.3 | 79.1±1.3 |
| BC-Z | 96.3±0.7 | 77.5±1.0 | 49.9±4.4 | 14.2±0.7 | 64.0±1.8 |
| GRIF | 96.6±0.8 | 89.4±2.5 | 87.6±0.1 | 27.7±3.7 | 78.6±2.5 |
| IQL | **98.2**±0.4 | 87.9±1.4 | 73.7±1.1 | 26.2±3.5 | 75.2±0.7 |
| CQL | **98.7**±0.3 | 92.2±0.9 | 83.8±1.7 | 25.6±2.5 | 78.1±1.6 |
| DAIL (ours) | 97.2±0.2 | **96.5**±1.4 | **94.9**±1.4 | **57.9**±0.9 | **89.2**±0.5 |
| *Out of Distribution* | | | | | |
| GCBC | 94.4±2.5 | 90.3±1.6 | 78.4±2.1 | 27.4±1.6 | 74.1±0.7 |
| BC-Z | 93.7±1.0 | 76.9±3.0 | 45.4±1.5 | 11.2±3.3 | 57.9±1.8 |
| GRIF | 95.9±1.7 | 88.8±2.6 | 75.6±3.9 | 22.5±2.7 | 71.2±2.6 |
| IQL | 98.0±0.4 | 86.1±1.2 | 70.4±3.6 | 21.4±3.1 | 69.7±2.3 |
| CQL | 98.8±0.5 | 88.9±2.1 | 71.9±2.2 | 27.6±0.8 | 72.6±0.4 |
| DAIL (ours) | **99.0**±0.2 | **91.3**±1.0 | **87.6**±2.0 | **49.1**±1.8 | **81.7**±1.3 |

**Results on the medium-quality dataset of BabyAI**   The success rates of in-distribution and out-of-distribution tasks on the medium-quality dataset are shown in Table 6. Due to the lower proportion of successful trajectories, learning in this dataset is more challenging. As a result, all methods show a significant decline in performance compared to results on the high-quality dataset (Table 1). However, our method still achieves optimal results, especially in the `PutNext` task category.

Table 6: Success rate of on the medium-quality dataset. Each score is evaluated over 3 seeds.

| Algorithm | In Distribution | | Out of Distribution | |
|---|---|---|---|---|
| | PutNext | All | PutNext | All |
| GCBC | 15.6±1.5 | 58.0±2.2 | 10.5±1.3 | 52.6±2.3 |
| BC-Z | 4.4±1.0 | 54.2±0.2 | 5.0±1.6 | 49.7±1.4 |
| GRIF | 7.0±1.5 | 61.4±0.2 | 4.9±2.5 | 54.7±0.6 |
| IQL | 17.7±2.8 | 67.3±0.5 | 12.3±0.9 | 61.4±0.0 |
| CQL | 13.5±1.8 | 69.2±0.7 | 14.5±1.9 | 63.4±1.3 |
| Ours | **32.8**±2.7 | **81.3**±0.7 | **26.4**±0.6 | **73.7**±0.6 |

**Ablation of components**   To study the contribution of each component in our learning framework, we conduct the following ablation study. We compare the performance of algorithms that only

Table 7: Ablation results of AUC on the high-quality dataset. Each score is evaluated over 3 seeds.

| Algorithm | 10 | 20 |
|---|---|---|
| CQL | 38,385.3±853.5 | 81,876.0±848.0 |
| DAIL w/o Alignment | 40,374.7±601.3 | 85,247.1±540.2 |
| DAIL w/o Distributional | 40,752.9±248.7 | 85,416.5±543.6 |
| DAIL | **42,370.6±404.8** | **88,450.5±291.6** |

apply trajectory-wise alignment or distributional language-guided policy alone with our method on `SynthLoc`. The experimental results in Figure 21 show that both modules significantly improve the performance over vanilla CQL on in-distribution and out-of-distribution tasks. Further, combining both components can achieve the best performance compared to other approaches.

We further evaluate the sample efficiency of each method by calculating the Area Under the Curve (AUC) for the success rates of their learning curves, and present the AUC comparisons of in-distribution learning curves at the key milestones of 10 and 20 epochs in Table 7. Statistical analysis confirms that at 20 epochs, DAIL holds a significant lead over the other three ablation models, as demonstrated by Kolmogorov-Smirnov tests on the AUC results (p = 0.004).

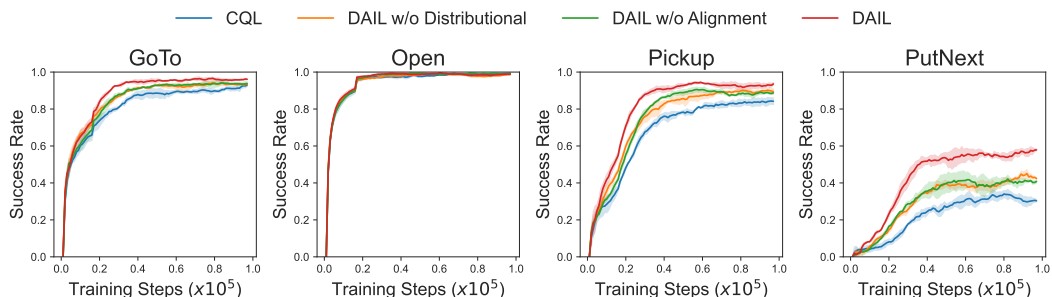

Figure 21: Ablation experiments on BabyAI tasks. The success rates are evaluated over 3 seeds.

**Quantitative measure of clustering.**  To quantitatively demonstrate the impact of different components in DAIL on representation clustering, we measure the clustering quality with the Silhouette score[52], and present the result in Table 8. The labels required to calculate the score are defined as follows: two instructions share the same label if they require the agent to perform the same action on the same kind of object with the same color as the final target object. The distance between language embeddings is measured using cosine distance.

The results demonstrate that in the All-task setting, the clustering metric of CQL even exhibits negative values, indicating pronounced task confusion. In contrast, both proposed methods evidently improve clustering performance.

Table 8: Silhouette score of in-distribution BabyAI tasks.

| Algorithm | All | PutNext |
|---|---|---|
| CQL | -0.030±0.005 | 0.004±0.007 |
| DAIL w/o Alignment | 0.024±0.016 | 0.048±0.008 |
| DAIL w/o Distributional | 0.110±0.019 | 0.088±0.020 |
| DAIL | **0.127±0.013** | **0.107±0.012** |

**Ablation of $\alpha$**  In Equation 13, $\alpha$ is the weight of the CQL loss. The ablation is done on the high-quality dataset in BabyAI tasks with various $\alpha$. The experimental results in Table 9 indicates that a wide range of $\alpha$ values (from 0.5 to 2) yield comparable performance, which drops off at the extremes ($\alpha = 0.2$ and $\alpha = 5$). We therefore recommend setting $\alpha$ between 0.5 and 2.

Table 9: Ablation experimental results on $\alpha$. Each score is evaluated over 3 seeds.

| $\alpha$ | In Distribution | | Out of Distribution | |
|---|---|---|---|---|
| | PutNext | All | PutNext | All |
| 0.2 | 44.0±3.6 | 83.7±0.7 | 40.8±5.1 | 78.4±1.7 |
| 0.5 | 57.4±2.2 | 88.1±0.3 | 50.7±2.7 | 83.1±0.8 |
| 1 | 56.2±3.6 | 87.7±1.3 | **50.8±3.7** | **84.1±0.7** |
| 2 | **57.9±0.9** | **89.2±0.5** | 49.1±1.8 | 81.7±1.3 |
| 5 | 44.8±3.2 | 85.2±0.2 | 37.1±5.1 | 77.4±1.7 |
| 10 | 35.7±2.3 | 82.8±1.1 | 39.1±4.4 | 77.2±0.7 |

# I  Demonstration Trajectories in ALFRED

We present extended trajectory visualizations of DAIL's task execution in the ALFRED benchmark, illustrating its semantic comprehension and generalization capabilities.

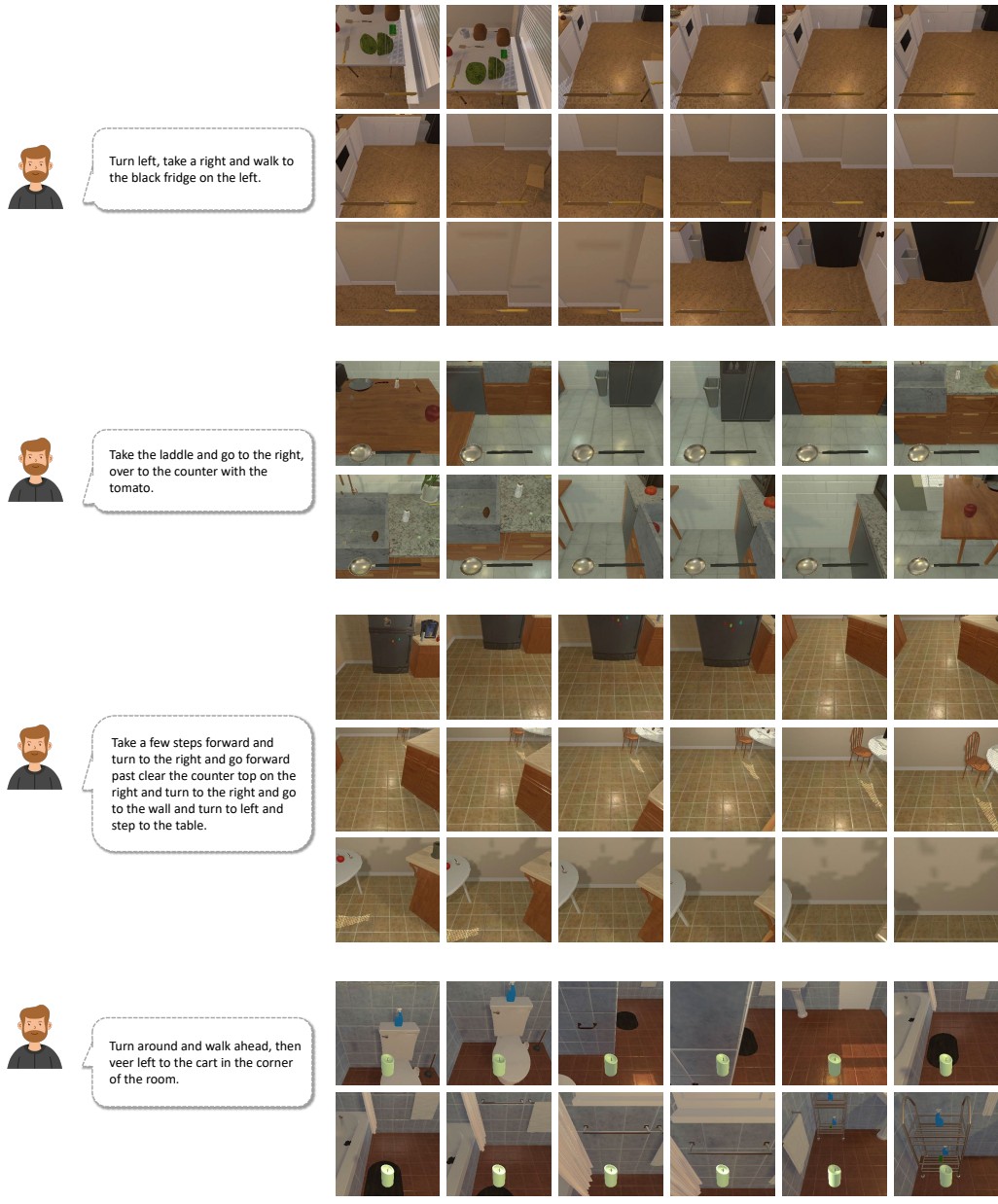

Figure 22: Extended trajectories of DAIL in ALFRED validation tasks.

