# OpenReview forum: "DAIL: Beyond Task Ambiguity for Language-Conditioned Reinforcement Learning"
_NeurIPS.cc/2025/Conference — NeurIPS 2025 poster_

### Official Review · Reviewer_eo9K · 2025-06-28

**Clarity:** 2
**Significance:** 3
**Originality:** 3
**Rating:** 5
**Confidence:** 3

**Summary:**

# Summary:

## Problem:

Natural Languages (NLs) is highly expressive but can sometimes be ambiguous. Current AIs struggle to maintain high performance when NL instructions for a given task are ambiguous. How to address this limitation?

## Contributions:

(i) Empirical analysis of the limitations of current mainstream methods in the face of NL-induced task ambiguity ;

(ii) Theoretical analysis of the relationship between task ambiguity and sample complexity ;

(ii) Present Distributional Aligned Learning (DAIL) agent, which leverages distributional RL to better map the relationships between complex observations, possibly-ambiguous language instructions, and reward signals. It also employs semantic alignment between the representations of trajectories and language instructions in order improve task discrimination by reducing the impact of ambiguous(/similar) NL instructions onto the policy learning process;

Evaluations on ALFRED and BabyAI benchmarks show that DAIL achieves high performance compared to baselines, even in visually complex settings (ALFRED).

**Questions:**

Please see Strengths and Weaknesses points for all question.
However, the two main actionable points that I would like to highlights are:

1. WQ3
2. WQ4
3. WC13

# POST-REBUTTAL :

Following the proposed changes and clarification from the authors' rebuttals, I am raising my scores:
- Quality: 1 -> 3
- Clarity: 1 -> 2
- Significance: 2 -> 3
- Originality: maintained at 3

I am also raising my overall score from 3 to 5.
I am waiting for subsequent authors-reviewers discussion before completely filling in my final justification.

**Ethical Concerns:**

["NO or VERY MINOR ethics concerns only"]

**Final Justification:**

Main issues resolved:

- WQ3 : lack of clarity was preventing me from properly evaluating the theoretical contributions. As far as I now understand, thanks to the authors' rebuttal clarifications, the quality of the theoretical contributions is high.

- WQ4: The paper was previously not including any quantitative measure of sample-efficiency whereas the paper was claiming improvements on that end. The authors' rebuttal provided statistically-significant results on that end. Thus, as far as I understand those results, the quality and significance of the paper has been raised by these additions.

- WC13: It was previously not clear what kind of generalisation improvement was being claimed. The authors' rebuttal have clarified that it is combinatorial generalisation (/systematicity). As I believe this kind of generalisation to be very important towards AGI, I think this clarification not only raises clarity but also the significance and quality of the paper, as the results present are subsequent (and likely statistically-significant).

I was assigning equal weights between those three main issues.

The rebuttal and discussions further highlighted the strenghts of the paper and addressed the major weaknesses, therefore I am recommending acceptance.

**Limitations:**

While Section 6 discusses 2 limitations pertaining to external validity of DAIL, and albeit vaguely, the paper does not discuss limitations of the experimental protocols and techniques it employs, principally in terms of internal validity (e.g. 'is the experimental protocol providing strong or weak evidence of the claims?' or 'is t-SNE the most appropriate approach to evidence the claim that DAIL's alignment improves task discrimination?' or 'why using t-SNE over other methods?').

The paper did not address potential negative societal impact of their work.

**Quality:**

3

**Strengths And Weaknesses:**

# Strengths & Weaknesses:

## Quality:

SQ0: I appreciate section 3’s formal definition and discussion and experiment, as it provides a solid problem framing.

WQ1: Regarding definition 1, I think it is important to further highlight to the reader that this definition is dependent on (1) a specific distrinction threshold $\delta$, and (2) a specific policy $\pi$ ; the latter feels like a possible limitation of the work and problem framing, which ought to be flagged so that it can later be improved upon.

WQ2: The experiment in Section 3 contains a few confounders that are not accounted for:

1. How does DAIL’s size compare to CQL’s ?

2. How would a CQL-half (or even -quarter) compare? I mean that model scaling might not require increase to resolve the challenge, but maybe rather decrease.


WQ3: As highlighted in WC7-11, the paper’s theoretical results are very difficult to comprehend due to lack of proper definition and notations. As the theoretical analysis is a very important contribution of the paper, I would advise the author to make the theorems and their proofs in the paper as detailed as possible, as there needs to be no doubt about their veracity for the paper to be possibly accepted in the conference. As it stands, I cannot recommend anything but rejection.

WQ4: Experimental protocol fails to emphasise measures of sample-efficiency. Indeed, only performance is being reported on in the form of success rates under a fixed budget of observations. While showing the learning curves provides some qualitative understanding of the sample efficiency of each method, it fails to provide a quantitative measure or, better yet, statistically-significantly-comparable measures about sample-efficiency. Thus, I would advise the authors to report the Area-Under-the-Curve (AuC) of the learning curves, for instance, as a quantitative measure of sample-efficiency, for starters. Then, in order to be as thorough as possible, I would advise the authors to perform statistical-significance tests on the results to provide stronger evidence. I believe this is a critical aspect of the paper’s quality, especially given that the theoretical improvement that the paper claims is along the line of sample-efficiency.

SQ5:Section 5.3 presents a very insightful experiment. However, I would advise the authors to consider adding a quantitative measure of clustering, if possible, in order to increase the strength of their evidence towards showing that DAIL’s Alignment is impacting the language instructions’ representations.

## Clarity:

WC1: ln24-26: the paper claims that MILLION[14] improves robot task acquisition in terms of (sample-)efficiency. However, MILLION[14] does not claim improvement in sample-efficiency but only in terms of success rate (performance). Thus, I think the second part of this sentence needs refactoring.

WC2: ln100 should probably point to Figure 2 (left). Then ln101 to middle.

WC3: ln116-122: this paragraph is repeating some sentences from ln35-42, it might be valuable to remove the repeated sentences to make for some space to address other issues. Moreover, this paragraph seems to be meant to work as an introduction for Section 4 but it fails to summarise the content of the subsections of Section 4.

WC5: ln126: ‘when the number of instructions increases’ is slightly misleading on its own. I would advise the author to also add: ‘… while keeping the number of actually-distinct tasks constant’ . By this, I mean to convey the nuance between learning difficulties due to the increase in size of the goal (and task) space from the learning difficulties due to the increased ambiguity between tasks, as the paper is focused on the latter only. Indeed, an increase in the number of instructions could be due to an increase in the number of actually-distinct tasks, which is not the context of the paper.

WC6: Section  4.1 employs many Bellman-related operators which have not been formally defined in the preliminaries (Section 2). Clarity and readability of the paper would be enhanced by providing their definitions and some motivating examples, possibly.

If not in the preliminaries section, at least in Appendix.

WC7: ln138: ‘we can discretize the value function distribution Z and train it by…’ : My understanding is that Z is the shorthand for $Z^\pi\in\mathbb{Z}$, which is the random variable that represents ‘the cumulative discounted reward obtained along the policy $\pi$’. Thus, I do not understand what the sentence mean when it says it, the random variable, can be trained.

WC8: In a similarly puzzling fashion, $\hat \theta$ is introduced in ln141 and equation 6 without definition. It seems to be used as the target of the optimisation of equation 6, but it is not explained how to obtain it in the first place.

WC9: ln552 describe an assumptions without justification and therefore is not rigorous enough for a proof. I advise the author to possibly introduce a formal lemma to address this assumption, after the vocabulary has been made more formal. Indeed, ‘can be regarded as’ is an informal expression whose formal meaning has not been detailed.

WC10: ln554 : ‘Following [5,29]’ is a very ambiguous reference. This is rather detrimental to the clarity and understandability of the proof, and therefore should probably be addressed with a lemma leading to equation 22 or equivalent. As it stands, I cannot evaluate the veracity of said proof due to WC9 and WC10.

WC11: ln553 : In a minor fashion, the terms ‘sub-gaussian’ and ‘1-Lipschitz’ have been used without definition. This hurts the clarity and readability of the proof and overall claim.

WC12: Figure 5’s right graphs (Pickup and PutNext) are using different y-axis scales compared to the left graphs, which makes it possibly misleading to a distracted reader. I would advise the author to set a fixed y-axis scale between 0 and 1.

WC13: ln206-209 highlights division of the task space between in-distribution and out-of-distribution, but it is not clear how the division is performed. Is it investigating combinatorial/systematic generalisation, i.e. is the division performed over instructions’ action-object pairs, e.g. ‘open red/blue door’ in the training set but ‘open yellow door’ is only found in the test set, and similarly with goto and pickup-based tasks, except with different colour being held out for the test set ?

WC14: What about the test-train split in the context of the ALFRED benchmark?

WC15: In equation 11, what is the meaning of ‘w.p.’ ? ‘z_i’ and ‘p_i’ are also left undefined. Please clarify.

## Significance:

SS1: The interplay between language instructions and behaviours is a critical topic, and I think that this paper makes both valuable theoretical and practical contributions to address this novel perspective on task ambiguity.

SS2: I appreciate the BabyAI test-train split being highlighted towards evaluating generalisation capabilities, as I think it is a valuable evaluation to make. However, the current amount of details about the test-train split does not allow me to evaluate what kind of generalisation is actually evaluated, as mentioned in WC13.

## Originality:

SO1: To my knowledge, this is the first paper to highlight the problem of task-ambiguity in the context of language-conditioned RL, and its formalisation of the problem is appreciated.

WO1: the related work paragraphs in the introduction could be improved to be more thorough, starting with, e.g. [Misra2017] for its original model that first attempted to ‘jointly reason about linguistic and visual input’, moreover within an reward-shaped RL context ; [Fried2018] for further characterisation of the difficulties caused by the flexibility of NLs, there in terms of NL instructions ‘identify[ing] only a few high-level decisions and landmarks rather than complete low-level motor behaviours’ ; [Co-Reyes2018] for proposing a trail-and-error-and-correction learning paradigm (going a step further than MILLION[14]) to deal with the ambiguities of NL instructions in language-conditioned RL, and actually claiming improved sample-efficiency.

WO2: I am now realising about Appendix A- Related Works. I would advise the author to reference the Appendix A from the introduction. Moreover, ln466 is an ambiguous statement without clear reference, I would advise the authors to rephrase and add a proper citation.

WO3: Semantic alignment of languages  with a contrastive learning approach is also investigated in a similar sparse-reward and language-conditioned RL context in [Denamganai2023]. The related works discussion (ln24-33) should probably include it and highlight how differently useful semantic alignment can be in RL contexts.

## Misc:

ln36 and ln119: ‘retains’ is an ambiguous term to be used here and I don’t think it unequivocally conveys the meaning that is intended. The whole sentence seems to want to emphasise how the distributional aspect of the RL module (policy and value function) allows for a more expressive representation, compared to non-distributional counterpart, and therefore should be more equipped to dealing with the ambiguities in the NL instructions. If I understand correctly, then I would advise a reformulation of this important sentence that introduces the specificity of DAIL. (ln37-39 is doing a far greater job driving the point home, though).

ln39: ‘constrains the language instructions’ --> I do not think I understand, as I do not think an input to the model, here the language instruction, can be constrained by the model. I assume that it is the ‘language instruction’ representation\* that is constrained?

ln36-42: The paragraph is constructed in a bottom-up approach, by going from the fine details about the policy and semantic alignment to the coarse claim of DAIL ‘improving learning efficiency’. As a neurodivergent individual, I appreciate this construction, but neurotypical readers might find it difficult to read. They might find more clarity through a top-down approach, starting with the coarse claim and then describing what the finer details that enable it.

ln43: It might be valuable to add a sentence that explicitly states why using structure vs visual observations (e.g. to check external validity of the DAIL agent, from structured to visual, and therefore from less to more complex and more expressive, observations).

ln95: ‘sematics’ -> ‘seman\*tics’

ln144: ‘denote the number\* of\* samples’ maybe ? Also, there might be a need to add a ‘respectively’ at the end of the sentence.

# References:

[Misra2017]: [Mapping Instructions and Visual Observations to Actions with Reinforcement Learning](https://aclanthology.org/D17-1106/) (Misra et al., EMNLP 2017)

[Fried2018]: Fried, Daniel, et al. "Speaker-follower models for vision-and-language navigation." Advances in neural information processing systems 31 (2018).

[Co-Reyes2018]: Co-Reyes, John D., et al. "Guiding Policies with Language via Meta-Learning." International Conference on Learning Representations. 2018.

[Denamganai2023]: Denamganaï, Kevin, et al. "ETHER: Aligning Emergent Communication for Hindsight Experience Replay." arXiv preprint arXiv:2307.15494 (2023).

---

> ### Author Rebuttal · Authors · 2025-07-31
>
> Dear Reviewer,
>
> Thanks for finding our paper novel, empirical rigor and practical relevance. We hope the following statement clears your concern.
>
> **WQ1: Emphasize the key points of Definition 1.**
>
> **A for WQ1:**
> - In the revised version, we have updated Definition 1 to emphasize predefined task distinction thresholds $\delta$ and $d$.
>
> - We have refined the requirement on the policy from "a specific policy $\pi$" to "any $\epsilon$-optimal policy $\pi$ that $V_{\pi}(s)\geq V^*(s)-\epsilon, \forall s \in \mathcal{S}$".
>
> ---
>
> **WQ2: A few confounders that are not accounted for.**
>
> **A for WQ2:**
>
> -  The sizes of these models in the toy experiment are as follows: CQL-double 56,075, CQL 23,947, DAIL 33,697. DAIL's size is between CQL and CQL-double.  When in BabyAI, the numbers of parameters are: CQL 1,481,217, DAIL 1,488,179, where the impact of this parameter scale becomes even less significant.
>
> -  As suggested, we run a CQL-half and a CQL-quarter on the toy experiment, and here are the results. Due to space constraints, we have presented only the results for 32, 128, and 512 tasks for reference (in percentage).
>
> **Table 1**: Average success rates over 100 evaluations for each number of instructions.
>
> |Model|32|128|512|
> |--|--|--|--|
> |CQL-quarter|91.2±5.8|86.0±2.1|70.9±0.9|
> |CQL-half|80.5±7.4|74.9±0.1|74.4±3.6|
> |CQL|89.9±3.2|77.4±1.1|70.6±2.3|
> |DAIL|**96.6±2.1**|**96.1±2.0**|**95.3±1.2**|
>
> Results above indicate that even when the model's parameter scale is halved or reduced to one-quarter, CQL's performance remains nearly identical, which demonstrates that the model size is not the key confounder of the performance differences between CQL and DAIL.
>
> ---
>
> **WQ3 (WC7-11)**
>
> **A for WQ3:**
>
> We have reconstructed part of the proof and updated the theoretical results accordingly. The new sample efficiency result is as follows:
>
> $$n_{\text{value}} \geq \frac{C_{\text{value}} \log(3M^2 / \eta)}{\delta^2}, n_{\text{dist}} \geq \frac{C_{\text{dist}} \log(3M^2 / \eta)}{d^2}$$
>
> To ensure clarity and coherence, we first provide a brief summary of the revised proof, followed by our responses to each of the comments.
>
> **Proofs:(modifying from line 552)**
>
> Following the triangle inequality of the absolute value, we have: **(WC9)**
> $$
> \left|\mathbb{E}\_{(s,a)\sim\mathcal{D}}[|\hat{Q}\_i-\hat{Q}\_j|]-\mathbb{E}\_{\pi}[|Q\_i-Q\_j|]\right|
> \leq \left|\mathbb{E}\_{(s,a)\sim\mathcal{D}}[|\hat{Q}\_i-\hat{Q}\_j|]-\mathbb{E}\_\mathcal{D}\mathbb{E}\_{(s,a)\sim\mathcal{D}}[|\hat{Q}\_i-\hat{Q}\_j|]\right|+ \left|\mathbb{E}\_\mathcal{D}\mathbb{E}\_{(s,a)\sim\mathcal{D}}[|\hat{Q}\_i-\hat{Q}\_j|]-\mathbb{E}\_{\pi}[|Q\_i-Q\_j|]\right|
> $$
>
> In general, we assume that the offline RL dataset is collected by an $\epsilon$-optimal policy, i.e., there exists an $\epsilon$-optimal policy $\pi$ that $\mathcal{D}$ satisfies
> $$
> \mathbb{E}\_\mathcal{D}\,\mathbb{E}\_{(s,a)\sim\mathcal{D}}[|\hat{Q}\_i-\hat{Q}\_j|]
> = \mathbb{E}\_{\pi}[|\hat{Q}\_i-\hat{Q}\_j|],
> \quad \mathbb{E}\_\mathcal{D}(\hat{Q}) = Q\_{\pi} = Q
> $$
>
> Based on the above derivation steps (with some intermediate steps omitted for word limit), we can upper-bound the original expression by following the three expressions,
> $\left|\mathbb{E}\_{(s,a)\sim\mathcal{D}}[|\hat{Q}\_i-\hat{Q}\_j|]-\mathbb{E}\_\mathcal{D}\mathbb{E}\_{(s,a)\sim\mathcal{D}}[|\hat{Q}\_i-\hat{Q}\_j|]\right|$ (term 1), $\mathbb{E}\_{\pi}|\hat{Q}\_i-Q\_i|$ (term 2), $\mathbb{E}\_{\pi}|\hat{Q}\_j-Q\_j|$ (term 3).
>
> Similar to the derivation of probabilities in Line 569 of the original text, we can also convert the original probability into the sum of the probabilities that each of the three terms exceeds
> $d/2$. Moreover, each of these three terms takes the form of an empirical estimate approximating the true expectation. Based on the existing conditions, we can apply Theorem 1 (Hoeffding's inequality) in[1], and obtain that for $\forall l_i,l_j\in \mathcal{L}$, **(WC10, 11)**
> $$
> \mathrm{Pr}\left(\text{term 1, 2, 3}\geq \delta/2\right)\leq 2\exp\left(-\tfrac{n\delta^2}{C\_{\text{value}}}\right)
> ...
> $$
>
> The subsequent proof follows the same reasoning as in the original manuscript. Besides, we modified the proof of distributional RL accordingly, where we upper-bound the original expression by similar three expressions.
>
> **Answers:**
>
> - *A for WC7:* Z is a shorthand of $Z^\pi$ and denotes a random variable. However, we often do not have access to the random variable Z but only its empirical version $Z_{\theta}$, which is a parameterized model to be learned.
>
> - *A for WC8:* Yes, $\hat{\theta}$ denotes the parameters of a target network, a common stabilization technique in deep RL. This network is initialized identically to the learning model but updated periodically (e.g., via polyak averaging
> $\hat{\theta}\leftarrow \tau \hat{\theta}+(1-\tau)\theta$). Its primary role is to provide stable regression targets for the Bellman equation.
>
> - *A for WC9:* We have removed the vague assumption in Line 552 and given a more rigorous derivation.
>
> - *A for WC10:* In the revised version, we have explicitly indicated the theorem from the referenced work. In addition, we use Corollary 5.2 and Remark 5 in [2] for proofs of distributional RL.
>
> - *A for WC11:* We have modified the proof to eliminate the dependence on arguments related to "sub-Gaussian" and "1-Lipschitz" properties.
> ---
>
> **WQ4: Quantitative measures of sample-efficiency.**
>
> **A for WQ4:**
>
> We have computed the AUC for the learning curves, and present the AUC comparisons of ID learning curves at the key milestones of 10 and 20 epochs below. The complete results are comprehensively detailed in the revised manuscript.
>
> **Table 2**: AUC of in-distribution BabyAI tasks.
>
> |Algorithm|10(↑)|20(↑)|
> |--|--|--|
> |CQL|38,385.3±853.5|81,876.0±848.0|
> |DAIL w/o Alignment|40,374.7±601.3|85,247.1±540.2|
> |DAIL w/o Distributional|40,752.9±248.7|85,416.5±543.6|
> |DAIL|**42,370.6±404.8**|**88,450.5±291.6**|
>
> **Statistical-significance test**. As suggested, we perform statistical significance tests on these AUC results at 20 epochs. The result shows that, at the $\alpha=0.05$ significance level, paired t-tests demonstrate statistically significant performance improvements for both the distributional component (p=0.0342) and the semantic alignment module (p=0.0332) individually. Furthermore, DAIL yields highly significant enhancements (p=0.002), confirming the synergistic efficacy of the complete methodology.
>
> These results of both epochs confirm the significant sample efficiency gains of our method.
>
> ---
>
> **SQ5: Quantitative measure of clustering**
>
> **A for SQ5:**
>
> As suggested, we measure the clustering quality with the Silhouette score[3]. The labels required to calculate the score are defined as follows: two instructions share the same label if they require the agent to perform the same action on the same kind of object with the same color as the final target object.
> The distance between language embeddings is measured using cosine distance.
>
> **Table 3**: Silhouette score of in-distribution BabyAI tasks.
>
> |Algorithm|All(↑)|PutNext(↑)|
> |--|--|--|
> |CQL|-0.030±0.005|0.004±0.007|
> |DAIL w/o Alignment|0.024±0.016|0.048±0.008|
> |DAIL w/o Distributional|0.110±0.019|0.088±0.020|
> |DAIL|**0.127±0.013**|**0.107±0.012**|
>
> The results demonstrate that in the All-task scenario, the clustering metric of CQL even exhibits negative values, indicating a severe task confusion phenomenon where it fails to effectively partition tasks, while both proposed methods evidently improve clustering performance.
>
> ---
>
> **WC1-WC6 and WC12: Clarity issues.**
>
> **A for WC1-WC6 and WC12:**
>
> We sincerely thank you for identifying the areas needing clarification or reference in our manuscript. These points have been carefully addressed in the revised version.
>
> **WC13, SS2: Test-train split in BabyAI**
>
> **A for WC13, SS2:**
>
> Our experimental design indeed aims to evaluate compositional generalization. We use only the full given instruction (verbatim text) as the unit for splitting. We sampled approximately 60\% of the unique instructions, and the training set contains only the trajectories in which the agent attempted to complete these tasks. We only constrain whether an instruction's text is included in the set. In Appendix E.2.1, we describe more details of ID/OOD split.
>
> **WC14: Test-train split in ALFRED**
>
> **A for WC14:**
>
> For the ALFRED dataset, we strictly adhere to the original work's train-test split between the Training set and Evaluation set (see Appendix E.2.2 for details). While this setup inherently tests generalization capabilities, it differs from BabyAI’s ID/OOD split: the original ALFRED split allows partial instruction overlap between training and evaluation sets (despite environmental variations).
>
> **WC15: Clarify ``w.p.'', $z_i$, and $p_i$ in Equation 11.**
>
> **A for WC15:**
>
> Equation 11 models the discrete value distribution $Z_\theta$ with two key components:
> - $z_i$ (termed *atoms* in our manuscript), which are support points on the quantized value domain.
> - $p_i$, representing the probability mass that the state-action value matches the support point $z_i$.
>
> "w.p." here stands for "with probability". Please refer to Appendix C for further explanations. We have revised this paragraph to make it clearer.
>
> ---
>
> **WO1-WO3, Misc:**
>
> **A for WO1-WO3:**
>
> We sincerely appreciate your detailed suggestions on related works, word choice, paragraph organization, and the logical presentation. They have been comprehensively addressed in the revised manuscript.
>
> Thanks again for your detailed and valuable comments. We sincerely hope our response has cleared your concerns, and we are looking forward to more discussions.
>
> ---
>
> **References:**
>
> [1] Concentration inequalities. 2003.
>
> [2] Convergence and concentration of empirical measures under Wasserstein distance in unbounded functional spaces. 2020.
>
> [3] Silhouettes: A Graphical Aid to the Interpretation and Validation of Cluster Analysis, 1987.

---

> > ### Comment · Reviewer_eo9K · 2025-08-04
> >
> > Dear Authors,
> > Thank you for your rebuttal.
> > I am very satisfied with your changes and clarifications and I will raise my score.
> > I am addressing below some further points.
> >
> > ### WQ4: Quantitative measures of sample-efficiency.
> >
> > I appreciate the proposed changes, but I think Kolmogorov-Smirnov test would be more adequate. Indeed, in my understanding, paired t-test only compares the means of distributions (which may not be befitting if the distributions are multi modal, like it could be the case in these kind of experiments), whereas it would be more valid to compare the location and shape of the distribution themselves (without assumptions about their nature), which is what the KS two-sample test enables (e.g. https://docs.scipy.org/doc/scipy/reference/generated/scipy.stats.ks_2samp.html).
> >
> >
> > ### SQ5: Quantitative measure of clustering
> >
> > Please consider reporting in a subsequent column of the related table the p-values of a KS test between current row distribution and the last-row DAIL distribution, in order to evidence how (statistically) significant each of the contributions are. I think it would make the Alignment addition shine even further, and therefore make your contribution even more valuable, especially in comparison to the (not-so-novel) distributional addition.

---

> > > ### Author Response · Authors · 2025-08-06
> > > **Thanks for raising the score!**
> > >
> > > Dear Reviewer,
> > >
> > > We would like to thank the reviewer for raising the score! We also appreciate the valuable comments, which helped us significantly improve the paper's strengths. We will address your follow-up points below.
> > >
> > > ---
> > >
> > > **A for WQ4:**
> > > Yes, the Kolmogorov-Smirnov test indeed better aligns with what we aim to verify. As suggested, we report the p-value of DAIL over ablation models from this test for AUC at epoch 20 as follows (5 seeds):
> > >
> > > **Table 1**: p-value of AUC from the Kolmogorov-Smirnov test.
> > >
> > > |Algorithm|All|PutNext|
> > > |--|--|--|
> > > |CQL|0.004|0.004|
> > > |DAIL w/o Alignment|0.004|0.004|
> > > |DAIL w/o Distributional|0.004|0.004|
> > > |DAIL|-|-|
> > >
> > > The improvements we propose led to such significant differences that the p-value of the KS test reaches a minimum value under the current number of seeds.
> > >
> > > ---
> > >
> > > **A for SQ5:**
> > > Thanks for your suggestion, which helps highlight our contribution more clearly. The results are as follows.
> > >
> > > **Table 2**: Silhouette score with p-value from the Kolmogorov-Smirnov test.
> > >
> > > |Algorithm|All(↑)|All p-value(↓)|PutNext(↑)|PutNext p-value(↓)|
> > > |--|--|--|--|--|
> > > |CQL|-0.030±0.005|0.004|0.004±0.007|0.004|
> > > |DAIL w/o Alignment|0.024±0.016|0.004|0.048±0.008|0.004|
> > > |DAIL w/o Distributional|0.110±0.019|0.040|0.088±0.020|0.040|
> > > |DAIL|**0.127±0.013**|-|**0.107±0.012**|-|
> > >
> > > We have added the clustering scores and the results of the KS test to the revised paper.
> > >
> > > ---
> > >
> > > Thanks again for your detailed and valuable comments.
> > > We sincerely hope our response has cleared your remaining concerns.
> > >
> > > Best,
> > >
> > > The Authors

---

### Official Review · Reviewer_SRtx · 2025-07-01

**Clarity:** 3
**Significance:** 2
**Originality:** 2
**Rating:** 4
**Confidence:** 3

**Summary:**

This paper proposes a novel method called DAIL to address the task ambiguity problem in language-conditioned reinforcement learning. DAIL integrates two key components: a distributional policy that enhances task discrimination by estimating full return distributions, and a semantic alignment module that optimizes task representations by maximizing mutual information between trajectories and language instructions. Extensive experiments conducted on both structured observation (BabyAI) and visual navigation (ALFRED) benchmarks validate DAIL's effectiveness, with visualization analyses revealing its ability to learn task representations and exhibit generalization to out-of-distribution tasks.

**Questions:**

1. The current experimental setup appears to inversely model the task ambiguity problem - while true task ambiguity involves multiple instructions mapping to a single task, the experiments conversely examine multiple tasks sharing identical instructions. This fundamental mismatch raises questions about whether the evaluation properly addresses the core challenge the paper aims to solve.
2. The similar clustering patterns observed between DAIL and its ablated variant (w/o Distributional) in Figure 7 may potentially undermine the claimed contribution of the distributional component.
3. The paper may lack ablation studies on the α coefficient in the CQL loss term?

**Ethical Concerns:**

["NO or VERY MINOR ethics concerns only"]

**Final Justification:**

Thank you for the author's reply. At present, I think the my main concern has been resolved, so I have raised the score.

**Limitations:**

Yes.

**Paper Formatting Concerns:**

No.

**Quality:**

2

**Strengths And Weaknesses:**

Strengths：
1. The paper is well-written, with a logical flow and accessible language that enhances readability.
2. It introduces and formally defines the task ambiguity problem in language-conditioned reinforcement learning, providing a fresh perspective on this challenge.
3. Empirical results support the claims—DAIL outperforms established baselines across all reported tasks.

Weaknesses：
1. The paper inadequately justifies the practical significance and broader implications of addressing the task ambiguity problem, failing to establish its real-world relevance or potential impact on the field.
2. The selected baseline methods are insufficient for rigorous comparison, as they: (a) fail to include state-of-the-art offline RL approaches, and (b) omit relevant multi-task/meta-RL methods (e.g., CSRO[1] and UNICORN[2]) that could better demonstrate the proposed method's advantages in handling task ambiguity.
3. A fundamental disconnect exists between the proposed technical solution and the core challenges stemming from task ambiguity. Moreover, the design of Trajectory-Wise Semantic Alignment lacks sufficient explanation and empirical validation.

[1] Gao, Yunkai, et al. "Context shift reduction for offline meta-reinforcement learning."
[2] Li, Lanqing, et al. "Towards an information theoretic framework of context-based offline meta-reinforcement learning."

---

> ### Author Rebuttal · Authors · 2025-07-31
>
> Dear Reviewer,
>
> Thanks for finding our work fresh and accessible. We hope the following statement clears your concern.
>
> ---
>
> **W1: Inadequately justifying the practical significance**
>
> **A for W1:**
> Language-conditioned RL holds significant practical importance. There is a growing expectation that a single agent should be able to accomplish various tasks specified by human instructions[1].
>
> We present that task ambiguity is a realistic problem in language-conditioned RL. It is mainly obstructed by the complexity and diversity of language instructions.
>
> To make our methods practical in the real world, we choose ALFRED as one of benchmarks, which includes sequential decision-making tasks involving household activities through language instructions and first-person vision. ALFRED is widely used for testing visual navigation and embodied agents[2], our experiments on ALFRED show high real-world relevance.
>
> ---
>
> **W2: Offline RL baselines and multi-task/meta-RL methods**
>
> **A for W2:**
>
> As suggested, we conduct extra experiments on state-of-the-art offline RL (MCQ[3], TD3+BC[4]) and meta RL methods (CCM[5], FOCAL[6], CSRO[7], UNICORN[8]). The experimental results of training from the high-quality dataset of BabyAI are as follows:
>
> **Table 1**: Performance of out-of-distribution BabyAI tasks.
>
> |   Algorithm   | Open | Goto | PickUp | PutNext | All |
> |:-------------:|:----:|:----:|:------:|:-------:|:---:|
> |   MCQ  | 91.9 ± 6.2 | 80.1±2.1 | 59.9±2.1 | 18.9±1.6 | 63.8±1.7 |
> |     TD3+BC    | 96.5±0.5 | 90.4±2.0 | 85.4±2.0 | 35.8±2.6 | 77.5±1.3 |
> |      CCM      | 98.6±1.2 | 87.5±1.3 | 68.0±4.2 | 26.1±4.5 | 69.5±0.5 |
> |     FOCAL     | 98.2±0.4 | 86.9±2.2 | 74.3±5.7 | 27.7±0.7 | 72.5±2.8 |
> |      CSRO     | 98.8±0.2 | 89.8±1.6 | 74.5±2.1 | 26.1±1.8 | 72.5±0.8 |
> |    UNICORN    | 99.0±0.5 | 88.0±3.6 | 73.3±4.6 | 29.9±1.2 | 73.5±2.4 |
>
> - Offline RL: The focus of offline RL methods is to avoid the phenomenon where policies perform poorly in unknown state-action spaces, which differs significantly from the task ambiguity problem addressed in this paper. Consequently, the performance gains prove ineffective or even detrimental.
>
> - Meta-RL: They focus on encoding trajectories to infer implicit task information. In our setting, however, task information is explicitly provided through instructions, and the real challenge lies in the vast number and diversity of instructions, as well as their complex relationships with tasks. Since the emphasis of meta-RL research does not address these challenges, its improvements are limited, and it is ineffective in resolving task ambiguity.
>
> ---
>
> **W3.1: Disconnect between the proposed technical solution and core challenges**
>
> **A for W3.1:**
>
> We respectfully disagree with your view that our method fails to address the core issue of task ambiguity, for reasons below:
>
> - **Definition**: In our work, task ambiguity refers to the phenomenon in language-conditioned tasks where agents confuse similar tasks due to the vast scale and fuzzy boundaries of instructions, leading to performance degradation. To address this, we tackle task ambiguity from both policy learning and representation learning perspectives.
> - **Policy Learning**: We are the first to theoretically demonstrate that distributional RL methods can effectively enhance an agent's ability to distinguish between tasks, and we apply this to language-conditioned problems.
> - **Representation Learning**: We propose a Trajectory-Wise Semantic Alignment, which leverages the correspondence between instructions and trajectories unique in language-conditioned tasks to apply contrastive learning, which directly enhances task discrimination in the language embedding space.
> - **Experimental Validation**: Our comprehensive experimental results demonstrate the effectiveness of the aforementioned methods. Furthermore, visualization experiments (Figure 7) qualitatively validate that our proposed method effectively solves the task ambiguity. Further, we quantitatively measure the improvement in embedding quality using the Silhouette score[9]. The results demonstrate that both proposed methods contribute to better language embeddings.
>
> **Table 2**: Silhouette score of task embedding on BabyAI.
>
> |Algorithm|All|PutNext|
> |--|--|--|
> |CQL|-0.030±0.005|0.004±0.007|
> |DAIL w/o Alignment|0.024±0.016|0.048±0.008|
> |DAIL w/o Distributional|0.110±0.019|0.088±0.020|
> |DAIL|**0.127±0.013**|**0.107±0.012**|
>
> **W3.2: Trajectory-Wise Semantic Alignment lacks explanation and empirical validation**
>
> **A for W3.2:**
>
> As discussed in **A for W3.1**, we design Trajectory-Wise Semantic Alignment to enhance the distinguishability of different tasks. To address your concern about the design of this component, we elaborate detailed implementations and experimental validations as follows:
>
> - Implementation: We utilize the correspondence between instructions and trajectories to implement contrastive learning. In the dataset, each trajectory is annotated with an instruction describing its behavior. For a given trajectory, we treat its annotated instruction as a positive sample and select any other instruction annotation from the dataset as a negative sample. The trajectory encoding is derived from the output of a sequence model encoding history information (Equation 8), which is then used alongside language embeddings for contrastive learning training. The implementation details are provided in Figure 13 in Appendix F.
>
> - Empirical validation: We further validate the component's efficacy through ablation studies in Sections 5.3 and 5.4: 1) Figure 7 visually demonstrates how alignment sharpens task representations. 2) Table 6 quantitatively shows significant performance gains, particularly on PutNext tasks: CQL (25.6±2.5) $\to$ DAIL w/o Distributional (39.6±0.6), and DAIL w/o Alignment (39.1±1.0) $\to$ DAIL (57.9±0.9). These results confirm the contribution of the proposed improvement. Additional data and analysis are available in Appendix H.
>
> ---
>
> **Q1: The fundamental mismatch in experimental setup**
>
> **A for Q1:**
>
> The reviewer’s understanding of task ambiguity aligns with ours, and our experiments have been specifically designed to model this issue. We elaborate on this from two perspectives: (i)relationships between tasks and instructions, and (ii)errors caused by task ambiguity.
>
> - The relationship between instructions and tasks is complex. In practice, multiple instructions may correspond to the same task (e.g., in ALFRED, a single trajectory can be described by three different instructions). This makes task ambiguity highly likely to occur, where different instructions are incorrectly treated as corresponding to the same task.
>
> - Our experiments are designed to address task ambiguity. We observe that baseline methods often fail to learn task embeddings correctly, thereby confusing distinct task objectives(e.g., encoding "pick up a red/blue ball" into similar embeddings). Our goal is to separate the representations of inherently different policies.
>
> As a supplement, in our formulation, the definition of a "task" is solely determined by the reward distribution $p(r|s, a, l)$, and is independent of environment configurations such as map layouts or different initial states.
>
> ---
>
> **Q2: Clustering patterns between DAIL and DAIL w/o Distributional**
>
> **A for Q2:**
>
> - To address this issue, we quantitatively measure the clustering quality with the Silhouette score[9]. We present the scores in *Answer for W3.1* (Table 2). Results indicate that distributional indeed improves task embedding.
>
> - Appendix H presents ablation study results on the distributional component, as shown in Table 6. DAIL significantly outperforms DAIL w/o Distributional, demonstrating the effectiveness of distributional component.
>
> - Moreover, in Figure 7 and the ablation experiment, the difference between CQL and DAIL w/o Alignment also stems from the distributional component. We can observe a substantial improvement in both task embeddings and experimental performance.
>
> From these comparisons, we conclude that the distributional design makes a noteworthy contribution to the task embedding.
>
> ---
>
> **Q3: Ablation studies on $\alpha$**
>
> **A for Q3:**
>
> As suggested, we do the ablation studies on $\alpha$ coefficient on BabyAI. We present the results for PutNext and All tasks here with complete results added to the revised manuscript.
>
> **Table 4**: Ablation results on $\alpha$.
>
> |$\alpha$|ID PutNext|ID All|OOD PutNext|OOD All|
> |--|--|--|--|--|
> |0.2|44.0±3.6|83.7±0.7|40.8±5.1|78.4±1.7|
> |0.5|57.4±2.2|88.1±0.3|50.7±2.7|83.1±0.8|
> |1|56.2±3.6|87.7±1.3|50.8±3.7|84.1±0.7|
> |2|57.9±0.9|89.2±0.5|49.1±1.8|81.7±1.3|
> |5|44.8±3.2|85.2±0.2|37.1±5.1|77.4±1.7|
> |10|35.7±2.3|82.8±1.1|39.1±4.4|77.2±0.7|
>
> The ablation results above show that DAIL is not very sensitive to the choice of $\alpha$ between 0.5 and 2. The performance begins to degrade when the influence of the loss is either too small($\alpha=0.2$) or too large($\alpha=5$).
>
> ---
>
> **References:**
>
> [1] Rt-2: Vision-language-action models transfer web knowledge to robotic control. CoRL. PMLR, 2023.
>
> [2] Context-aware planning and environment-aware memory for instruction following embodied agents. ICCV, 2023.
>
> [3] Mildly conservative q-learning for offline reinforcement learning, NeurIPS 2022.
>
> [4] A minimalist approach to offline reinforcement learning, NeurIPS 2021.
>
> [5] Towards Effective Context for Meta-Reinforcement Learning: an Approach based on Contrastive Learning, AAAI 2020.
>
> [6] FOCAL: Efficient Fully-Offline Meta-Reinforcement Learning via Distance Metric Learning and Behavior Regularization, ICLR 2021.
>
> [7] Context shift reduction for offline meta-reinforcement learning, NeurIPS 2023.
>
> [8] Towards an information theoretic framework of context-based offline meta-reinforcement learning, NeurIPS 2024.
>
> [9] Silhouettes: A Graphical Aid to the Interpretation and Validation of Cluster Analysis. Journal of computational and applied mathematics, 1987.

---

> > ### Author Response · Authors · 2025-08-06
> > **Looking forward to further discussions!**
> >
> > Dear reviewer,
> >
> > We were wondering if our response and revision have cleared all your concerns. In the previous responses, we have tried to address all the points you have raised. In the remaining days of the rebuttal period, we would appreciate it if you could kindly let us know whether you have any other questions, so that we can still have time to respond and address. We are looking forward to discussions that can further improve our current manuscript. Thanks!
> >
> > Best regards,
> >
> > The Authors

---

> > ### Comment · Reviewer_SRtx · 2025-08-08
> >
> > Thank you for the author's reply. My concern has been resolved.

---

> > > ### Author Response · Authors · 2025-08-08
> > > **Glad to know your concern has been resolved.**
> > >
> > > Dear Reviewer,
> > >
> > > We are delighted to know that our rebuttal and the corresponding revisions have adequately addressed your concerns. We are very grateful for your time and insightful comments during this process. With your concerns now addressed, we sincerely hope that our work will earn your support in the final evaluation.
> > >
> > > Best regards,
> > >
> > > The Authors

---

### Official Review · Reviewer_ZEoJ · 2025-07-02

**Clarity:** 3
**Significance:** 2
**Originality:** 3
**Rating:** 4
**Confidence:** 3

**Summary:**

The paper proposes a better algorithm for language-conditioned offline RL tasks. It identifies one key challenge in such kind of RL tasks, that is to handle the vast space of language instructions and potential ambiguity in its relation to the final goal. To address this issue, the proposed method DIAL build upon CQL with two additional change
1. distributed value function to handle the ambiguity more directly.
2. cross similarity to enhance the representation.
The experiment shows better performance than baselines with the two addons.

**Questions:**

1. The difference of GoTo and Open can be differs if one use some exisitng pre-trained models like (RT-1). It is unclear if we use pre-trained components that it will solve the ambiguity problem directly.

**Ethical Concerns:**

["NO or VERY MINOR ethics concerns only"]

**Limitations:**

yes

**Paper Formatting Concerns:**

No issues.

**Quality:**

3

**Strengths And Weaknesses:**

Strength:
1. The proposed new loss are simple and easy to understand.
2. The experiments is conducted on a large set of benchmarks set which is comprehensive.

Weakness.
1. The proposed losses can be found origin from existing literature (distributed RL, and contrastive learning for similarity metrics), which reduces the novelty.
2. I don't understand the confusion of the language embedding relates to the final outcome of the policy. The t-sne is a lower dimension space, the points can still be different in high dimension. Moreover, go to door and open door shares a lot in common in going to the door first, which should have similar embedding. To my knowledge, as long as the Q is correct, it should learn each task individually. Does the benefit comes from distributed value itself (like in single-task RL policy literacture) rather than the latent embedding?

---

> ### Author Rebuttal · Authors · 2025-07-31
>
> Dear Reviewer,
>
> Thanks for finding our paper easy to understand and empirically comprehensive. The points you raised are explained in the following.
>
> ---
>
> **W1: Proposed losses can be found origin from the existing literature**
>
> **A for W1**:
>
> We appreciate your perspective that proposed losses can be found originating from existing literature. While we agree with this perspective, we wish to emphasize that our core contribution lies not in refining specific losses but rather in identifying critical challenges inherent to language instruction tasks and proposing targeted solutions to address them.
>
> To the best of our knowledge, this work is the first to focus on the task ambiguity problem inherent in language instruction learning. We clearly define this problem within the language-conditioned RL framework and empirically demonstrate its prevalence in existing RL methods through experimental analysis (Figure 2 and the results for IQL and CQL in Table 1).
>
> To address this problem, we attempt to tackle it from two perspectives: policy learning and representation learning.
> - For policy learning, we prove, for the first time, that distributional reinforcement learning methods can effectively enhance the agent's ability to distinguish tasks, and apply this approach to language-conditioned tasks.
> - For representation learning, we propose Trajectory-Wise Alignment, which leverages the correspondence between language instructions and trajectories unique to language-conditioned tasks. By introducing self-supervised constraints, this approach directly enhances the distinguishability of different tasks in the language embedding.
>
> Comprehensive experiments validate the substantial performance gains achieved by DAIL (Table 6 and Figure 20). These significant gains also underscore the critical importance of addressing the task ambiguity problem as we have claimed.
>
> ---
>
> **W2.1: Confusion of the language embedding relates to policy**
>
> **A for W2.1**:
>
> We claim that the confusion of language embeddings is closely related to the policy for the following reasons.
>
> Language embeddings provide the only information to distinguish between different tasks in Language-conditioned MDP settings. Therefore, when two distinct instructions are mapped to similar language embeddings, the agent struggles to differentiate between them, potentially leading to erroneous behaviors deviating from the original instructions.
>
> On the other hand, when language embeddings are similar, policy behaviors should also exhibit similarity in robust control systems (so-called Lipschitz continuity[1]). As a result, when embeddings exhibit ambiguity, this confusion propagates into the agent's behavioral patterns.
>
> Besides, our ablation studies (Table 6 and Figure 20) also support our claim that policies with more discernible language embeddings have better performances.
>
> **W2.2: The points can still be different in high dimensions**
>
> **A for W2.2**:
>
> To address your concern, we quantitatively measure the clustering quality with the Silhouette score[2] using language embeddings before dimensionality reduction. We present the scores for all tasks and PutNext sub-tasks below.
>
> **Table 1**: Silhouette score of in-distribution BabyAI tasks.
>
> |Algorithm|All(↑)|PutNext(↑)|
> |--|--|--|
> |CQL|-0.030±0.005|0.004±0.007|
> |DAIL w/o Alignment|0.024±0.016|0.048±0.008|
> |DAIL w/o Distributional|0.110±0.019|0.088±0.020|
> |DAIL|**0.127±0.013**|**0.107±0.012**|
>
> The results demonstrate that both proposed improvements contribute to enhancing the performance of language embeddings in high dimensions.
>
> **W2.3: Similar behaviors should have similar embeddings**
>
> **A for W2.3**:
>
> Yes, we agree. The results of our visualization also match your intuition. For example, in the third figure of Figure 7, 'Goto the door' is significantly closer to "Open the door" rather than to 'Goto the ball' since the former two share a lot more in behaviors.
>
> **W2.4: Does the benefit come from the distributed value itself rather than the latent embedding?**
>
> **A for W2.4**:
>
> Our partial experimental results support that the advantages of distributional RL are not solely due to the distributional value itself:
> - In the toy experiment, when the number of tasks is low, the performance gap between the distributional method and CQL is not substantial. However, as the number of tasks increases, its performance advantage becomes pronounced. This indicates that its benefits extend beyond the distributional value itself.
> - Comparing the results of CQL and DAIL w/o Alignment shown in Figure 7 (where the latter figure represents results applying the distributional method on top of CQL), the distributional method clearly enhances the discriminative power of the language embeddings. This improvement is particularly evident in tasks within the "Pickup/Goto the box" and "Pickup/Goto the key" groups, where tasks involving key versus box are distinctly separable."
>
> ---
>
> **Q1: Would pre-trained components solve the problem directly?**
>
> **A for Q1**:
>
> While pre-trained components can alleviate this issue, they cannot directly solve it. To illustrate this claim, we employ a more recent language encoder [3] (CLIP+LLM2CLIP version), which leverages large language models to enhance the training of linguistic encoders, thereby achieving superior capabilities in language understanding and discrimination. Using this encoder, we retrain our method along with CQL on the BabyAI high-quality dataset. Notably, we introduce an additional baseline, CQL-LLM2CLIP, which directly feeds the LLM2CLIP-embeddings as language embeddings into CQL's policy network without fine-tuning the language encoder. The experimental results are as follows:
>
> **Table 2**: Success rate of BabyAI tasks with LLM2CLIP.
>
> |Algorithm|ID PutNext|ID All|OOD PutNext|OOD ALL|
> |--|--|--|--|--|
> |CQL|25.9±2.1|78.8±0.9|22.8±2.3|70.2±0.5|
> |CQL-LLM2CLIP|36.2±1.2|81.5±0.5|28.7±2.7|70.3±1.4|
> |DAIL(ours)|**58.7±1.5**|**89.1±0.6**|**55.7±1.5**|**84.7±0.6**|
>
> The experimental results demonstrate that:
>
> - Adopting a stronger pre-trained component shows measurable gains for both CQL-LLM2CLIP and DAIL, confirming that a better pre-trained component can help alleviate the ambiguity problem.
>
> - The CQL vs. CQL-LLM2CLIP comparison demonstrates that despite possessing a powerful pre-trained component, CQL fails to effectively leverage these pre-trained embeddings, with its performance slightly underperforming even the non-finetuned version.
>
> All the above experimental results substantiate our claim that pre-trained components cannot fully resolve the ambiguity problem, emphasizing the critical importance of effectively learning frameworks, precisely the focus of our work.
>
> Thanks again for the valuable comments. We hope our response has cleared your concern. We are looking forward to more discussions.
>
> ---
>
> **References:**
>
> [1] Evaluating the robustness of neural networks: An extreme value theory approach[J]. ICLR, 2018.
>
> [2] Silhouettes: A Graphical Aid to the Interpretation and Validation of Cluster Analysis, Journal of computational and applied mathematics, 1987.
>
> [3] https://github.com/microsoft/LLM2CLIP

---

> > ### Author Response · Authors · 2025-08-06
> > **Looking forward to further discussions!**
> >
> > Dear reviewer,
> >
> > We were wondering if our response and revision have cleared all your concerns. In the previous responses, we have tried to address all the points you have raised. In the remaining days of the rebuttal period, we would appreciate it if you could kindly let us know whether you have any other questions, so that we can still have time to respond and address. We are looking forward to discussions that can further improve our current manuscript. Thanks!
> >
> > Best regards,
> >
> > The Authors

---

> > > ### Comment · Reviewer_ZEoJ · 2025-08-08
> > > **Thanks for the clarification.**
> > >
> > > Thanks for the additional results.

---

### Official Review · Reviewer_gNoU · 2025-07-02

**Clarity:** 2
**Significance:** 2
**Originality:** 2
**Rating:** 3
**Confidence:** 4

**Summary:**

The paper addresses the task ambiguity problem in language-conditioned RL tasks: proximity in the language instruction space does not necessarily imply task similarity. It introduces DAIL, a method that combines distributional RL with trajectory-based contrastive learning of task representations. They show the efficacy of their method through the BabyAI and ALFRED benchmark tasks.

**Questions:**

1. How does this work fit into the prior work in RL that studies learning task representations?
2. Could the authors provide a justification for Definition 1?
3. How tight are the sample complexity upper bounds? Are gains only in constants inside the log sufficient? Could one use a different concentration inequality to arrive at better constants?

**Ethical Concerns:**

["NO or VERY MINOR ethics concerns only"]

**Final Justification:**

The authors clarified the positioning of their work during the rebuttal.

**Limitations:**

Yes.

**Paper Formatting Concerns:**

None.

**Quality:**

2

**Strengths And Weaknesses:**

**Strengths**
- Task ambiguity is a problem very relevant for the RL community.

**Weaknesses**
- Language-conditioned RL is an instantiation of the contextual MDP setting [1], and prior work has studied learning useful task representations in this setting already [2, 3, 4, 5]. Surprisingly, this paper conveniently ignores these works and does not reference them at all.
- The trajectory-based contrastive learning of task representations is already well studied in the community [2, 3, 4, 5].
- Definition 1 is stated without sufficient justification. Why should similar tasks have similar Q-values?
- It is unclear how the authors draw the conclusion that distributional RL achieves better sample complexity for task disambiguation than value-based RL. Since the gains are not in the exponents but only in the constants, how do the authors arrive at the conclusion? The relation between C_value and C_dist should dictate the final conclusion.
- Overall, the method appears to be incremental and as such does not meet the standards for NeurIPS.

[1] Hallak et al. Contextual Markov Decision Processes, arXiv, 2015.

[2] Fu et al. Towards Effective Context for Meta-Reinforcement Learning: an Approach based on Contrastive Learning, AAAI 2020.

[3] Agarwal et al. Contrastive Behavioral Similarity Embeddings for Generalization in Reinforcement Learning, ICLR 2021.

[4] Li et al. FOCAL: Efficient Fully-Offline Meta-Reinforcement Learning via Distance Metric Learning and Behavior Regularization, ICLR 2021.

[5] Mahajan et al. Learning Embeddings for Sequential Tasks Using Population of Agents, IJCAI 2024.

---

> ### Author Rebuttal · Authors · 2025-07-31
>
> Dear Reviewer,
>
> Thank you for your valuable comments. We address the points you raised in the following, and we hope our responses can resolve your concerns.
>
> ---
>
> **W1, W2, and Q1: Lack of references to and comparisons with existing research on task representations in RL.**
>
> **A for W1, W2, and Q1:**
>
> Thanks for pointing out the relevant papers—these references are indeed valuable. We have studied all and reproduced several of the papers as the reviewer suggested, and our analysis and experimental results are summarized below:
>
> - **LCMDP**: While language-conditioned MDP can be viewed as a special case of contextual MDP, the key challenge differs a lot. LCMDP is mainly obstructed by the complexity and diversity of language instructions, which makes it difficult for the agent to ground them to the intended task goals correctly.
>
> - **Related works:** In our original paper, the Related Work section (Appendix A) focused on task representation of goal-conditioned reinforcement learning.
> We incorporate an extra discussion of the papers and related research that the reviewer mentioned in the revised version.
>
> - **Supplementary experiments：** Some papers the reviewer mentioned have a different research focus from our paper. They contribute to learning implicit task information from trajectories, while task information is explicitly provided through instructions in language-conditioned RL. As a result, the improvements of these algorithms are minor, and some cannot even be reproduced due to the experimental setting. We provide the extra experimental results and analysis as follows:
>
> **Table 1**: Performance of in-distribution BabyAI tasks.
>
> |   Algorithm   | Open | Goto | PickUp | PutNext | All |
> |:-------------:|:----:|:----:|:------:|:-------:|:---:|
> |      CCM      | 94.9±4.5 | 88.3±2.6 | 83.0±2.4 | 29.9±1.1 | 78.4±1.8 |
> |     FOCAL     | 98.6±0.3 | 90.5±2.6| 87.5±1.6  |  37.7±0.9 |81.8±0.8  |
> |      CSRO     | 98.0±0.5|93.1±1.6 | 87.6±1.5 | 36.7±1.7 | 82.2±1.5 |
> |    UNICORN    | 97.3±0.2|90.4±0.4|84.4±1.8|35.3±0.7|80.3±0.7|
>
> **Table 2**: Performance of out-of-distribution BabyAI tasks.
>
> |   Algorithm   | Open | Goto | PickUp | PutNext | All |
> |:-------------:|:----:|:----:|:------:|:-------:|:---:|
> |      CCM      | 98.6±1.2 | 87.5±1.3 | 68.0±4.2 | 26.1±4.5 | 69.5±0.5 |
> |     FOCAL     | 98.2±0.4 | 86.9±2.2 | 74.3±5.7| 27.7±0.7|72.5±2.8|
> |      CSRO     | 98.8±0.2 | 89.8±1.6 | 74.5±2.1 | 26.1±1.8 | 72.5±0.8 |
> |    UNICORN    | 99.0±0.5 | 88.0±3.6 | 73.3±4.6 | 29.9±1.2 | 73.5±2.4 |
>
> - CCM[1]: It introduces contrastive learning methods to enhance transition encoding, and we adapt it to language instruction encoding. The results indicate that this method provides only marginal improvement over CQL, primarily because it overlooks the relationship between language instructions and trajectories.
>
> - PSM[2]: This paper focuses on optimal policy transfer, which relies on learning state representations in similar scenarios and is limited to a single or a small number of tasks. In contrast, our work considers a partially observable environment that varies across runs, with diverse and abundant tasks. As a result, it is not applicable to our setting.
>
> - FOCAL[3]: It introduces distance metrics to separate different trajectory embeddings. We adapt it to language embedding, which shows some improvements on CQL. However, it still fails to account for the relationship between language instructions and trajectories, and thus does not fundamentally resolve the problem of task ambiguity.
>
> - [4]: This paper leverages the idea that two tasks are similar if observing an agent’s performance on one task reduces the uncertainty about its performance on the other. We did not implement it because the task similarity estimate needs rollouts from the environment, which is not available in offline settings.
>
> - CSRO[5] and UNICORN[6]: We adapt the task representation methods from two state-of-the-art meta-RL approaches to language-conditioned setting and conduct reproduction experiments. Their performance was comparable to that of FOCAL.
>
> ---
>
> **W3 and Q2: Definition 1 is stated without sufficient justification.**
>
> **A for W3 and Q2:**
>
> For language-conditioned policy $\pi(a|s,l)$, when task representations are similar, the corresponding policies should also exhibit similar behaviors. This not only helps prevent abrupt behavioral shifts but is also an important condition for designing robust control systems[8]. Conversely, if two different tasks are mapped to nearby positions in the task representation space, the agent may fail to distinguish between them, potentially resulting in incorrect behaviors.
>
> Moreover, given the complexity and vast number of language instructions, estimation errors in the Q-function can easily propagate backward and cause confusion in the task embeddings. To quantify this phenomenon, we introduce Definition 1.
>
>
> Consequently, in Definition 1, the "similarity" between tasks specifically refers to the similarity of their true Q-values.
>
> ---
>
> **W4: Theoretical analysis of sample complexity upper bounds.**
>
> **A for W4:**
>
> Although the sample efficiency gains are not in the exponents, our main focus is on the difference between the predefined task discrimination threshold of value-based ($\delta$) and distributional-based RL ($d$). Following the proof in Appendix D from our paper, we obtain that $\delta \leq d$, when the mean difference of the Q function is small but the distribution difference is large, $\delta<<d$.
>
> We also prove that both $C_{\text{value}}, C_{\text{dist}}\in (0, \mathcal{O}(Q_{\max}^2)]$ are nearly identical in terms of magnitude. In particular, since $Q_{\max}=1$ in most language-conditioned scenarios, their impact on the overall sample efficiency becomes even less.
>
> **Q3: Theoretical problems**
>
> **A for Q3:**
>
> - **Theoretical analysis:** The sample complexity upper bounds are quite tight. Our gains are not in constants inside the log sufficient but in the denominator ($\delta$ and $d$).
> We believe that even others use different concentration inequalities to achieve better bounds, they can only achieve smaller constant coefficients [9], which have little effect while the index of other variables remains the same.
>
> - **Experiments:** Moreover, we have computed the AUC(Area-Under-the-Curve) for the learning curves and statistical significance tests on these AUC results at 20 epochs. We present the AUC comparisons of ID learning curves at the key milestones of 10 and 20 epochs below.
>
>     **Table 3**: AUC of in-distribution BabyAI tasks. Each score is evaluated over 3 seeds.
>
>     |Algorithm|10(↑)|20(↑)|
>     |--|--|--|
>     |CQL|38,385.3±853.5|81,876.0±848.0|
>     |DAIL w/o Alignment|40,374.7±601.3|85,247.1±540.2|
>     |DAIL w/o Distributional|40,752.9±248.7|85,416.5±543.6|
>     |DAIL|**42,370.6±404.8**|**88,450.5±291.6**|
>
>     The statistical significance tests show that, at the $\alpha=0.05$ significance level, paired t-tests demonstrate statistically significant performance improvements for the distributional component (p=0.0342). Furthermore, DAIL yields highly significant enhancements (p=0.002). The above results indicate that the distributional component indeed contributes to achieving higher sample efficiency.
>
> ---
>
> **W5: The method appears to be incremental.**
>
> **A for W5:**
>
> We respectfully disagree that our method appears to be incremental for the following reasons:
>
> - **Theory**. To the best of our knowledge, this work is the first to focus on the task ambiguity problem inherent in language instruction learning. We clearly define this problem within the language-conditioned RL framework and empirically demonstrate its prevalence in existing RL methods through experimental analysis (Figure 2 and the results for IQL and CQL in Table 1).
>
> - **Method**. To address this problem, we attempt to tackle it from two perspectives: policy learning and representation learning.
>   * For policy learning, we prove, for the first time, that distributional reinforcement learning methods can effectively enhance the agent's ability to distinguish tasks, and apply this approach to language-conditioned tasks.
>   * For representation learning, we propose Trajectory-Wise Alignment, which leverages the correspondence between language instructions and trajectories unique to language-conditioned tasks. By introducing self-supervised constraints, this approach directly enhances the distinguishability of different tasks in the language embedding space.
>
> - **Experiment results**. Comprehensive experiments validate the substantial performance gains achieved by DAIL (Table 6 and Figure 20). These significant gains also underscore the critical importance of addressing the task ambiguity problem in LCMDP, as we have claimed.
>
> Thanks again for your timely and valuable comments. We sincerely hope that our response has cleared the concern you had regarding our problem settings and novelty, and we are looking forward to more discussions.
>
> ---
>
> **Reference:**
>
> [1] Fu et al. Towards Effective Context for Meta-Reinforcement Learning: an Approach based on Contrastive Learning, AAAI 2020.
>
> [2] Agarwal et al. Contrastive Behavioral Similarity Embeddings for Generalization in Reinforcement Learning, ICLR 2021.
>
> [3] Li et al. FOCAL: Efficient Fully-Offline Meta-Reinforcement Learning via Distance Metric Learning and Behavior Regularization, ICLR 2021.
>
> [4] Mahajan et al. Learning Embeddings for Sequential Tasks Using Population of Agents, IJCAI 2024.
>
> [5] Gao, Yunkai, et al. Context shift reduction for offline meta-reinforcement learning, NeurIPS 2023.
>
> [6] Li, Lanqing, et al. Towards an information theoretic framework of context-based offline meta-reinforcement learning, NeurIPS 2024.
>
> [7] Weng T W, Zhang H, Chen P Y, et al. Evaluating the robustness of neural networks: An extreme value theory approach. ICLR 2018.
>
> [8] Boucheron, et al. Concentration inequalities. 2003. 208-240.

---

> > ### Author Response · Authors · 2025-08-06
> > **Looking forward to further discussions!**
> >
> > Dear reviewer,
> >
> > We were wondering if our response and revision have cleared all your concerns. In the previous responses, we have tried to address all the points you have raised. In the remaining days of the rebuttal period, we would appreciate it if you could kindly let us know whether you have any other questions, so that we can still have time to respond and address. We are looking forward to discussions that can further improve our current manuscript. Thanks!
> >
> > Best regards,
> >
> > The Authors

---

> ### Comment · Reviewer_gNoU · 2025-08-07
>
> I thank the authors for their rebuttal.
>
> **W1, W2, and Q1**:  Thanks for the clarification.
>
> **W3 and Q2**: I'm not still not convinced with the justification. Distinction is defined w.r.t. a shared policy \pi. Now, this policy could be anything. Consider two tasks that have the same underlying MDP, but different observations: the first one is grayscale, the second is colored. Now, consider a policy \pi that solves the first task, but not the second. The Q values are going to be different, but the semantics match.
>
> **W4**: Having some gain (in terms of a constant multiplier) inside the log is equivalent to having a constant gain outside of it. I'm still not convinced by the argument.
>
> Based on the above points, I have increased my score to a 3.

---

> ### Author Response · Authors · 2025-08-07
> **Thanks for raising the score!**
>
> Dear Reviewer,
>
> We would like to thank the reviewer for raising the score! We also appreciate your further comments, and hope our following responses can resolve your remaining concerns.
>
> ---
>
> **A for W3 and Q2:**
>
> We address your concerns from the following two perspectives:
>
> First, we agree that relying on a specific policy is not rigorous enough. Following your suggestion, we have revised **"a specific policy $\pi$"** to **"any $\epsilon$-optimal policy $\pi$ such that $V_{\pi}(s)\geq V^*(s)-\epsilon,\ \forall s \in \mathcal{S}$"**. This modification makes the definition more general and facilitates a cleaner derivation.
>
> Second, in our setting, task embedding focuses solely on differences in reward functions, assuming a fixed observation space.
> When both colored and grayscale observations are present in the dataset, two "tasks" with the same semantics share identical rest of the conditions, so their optimal policies with coupled observations (and thus the corresponding $Q^*$) should be equivalent. In our revised version, we restrict Definition 1 to **any $\epsilon$-optimal policy** rather than a **specific policy**. Any such policy is capable of solving the task within a minor suboptimality. Consequently, the corresponding $Q_\pi$ are expected to be very similar, and we treat them as the same task under the definition.
>
> ---
>
> **A for W4:**
>
> We believe the advantage of the distributional component primarily lies in its effect on the denominator, i.e, the predefined task discrimination threshold of value-based ($\delta$) and distributional-based RL ($d$). In Appendix D, we prove that the absolute difference between the means of two distributions is upper bounded by their Wasserstein distance, i.e, $W_1(Z\_i,Z\_j)\geq |\mathbb{E}\_{Z_i}(Q)-\mathbb{E}\_{Z\_j}(Q)|$ ($Z\_i,Z\_j$ denotes distribution of Q values for task $i$ and $j$). Therefore, under the distributional setting, we can always define a finer (i.e., bigger) discrimination threshold, $\delta \leq d$.
> When the mean difference of the Q function is small but the distribution difference is large, we can have $\delta<<d$.
>
> Here is an intuitive example. Consider a multi-modal distribution $Z_1$ that takes the value 0.3 with 50\% probability and 0.1 with 50\% probability, giving it an expected value of 0.2. In contrast, distribution $Z_2$ always takes the fixed value 0.2. For non-distributional methods, the expectations of both distributions are 0.2, making it impossible to distinguish between them, whereas distributional methods can.
>
> The above example is highly relevant to our setting, while significant task ambiguity exists in language-conditioned tasks. Therefore, as with $Z_1$ and $Z_2$ in the example above, non-distributional methods struggle to distinguish between different tasks, which is supported by our results in AUC (measuring sample efficiency) shown in Table 3 in the previous rebuttal, with paired t-tests demonstrate statistically significant performance improvements for the distributional component (p=0.0342) at the $\alpha=0.05$ significance level.
>
> ---
>
> Thanks again for your valuable comments. We sincerely hope our response has cleared your remaining concerns.
>
> Best regards,
>
> The Authors

---

### Note · Authors · 2025-08-12

Dear Area Chair and Reviewers,

We are sincerely grateful to the Area Chair and all reviewers for their diligent work and thoughtful engagement, which is invaluable in strengthening our manuscript.

We are delighted that the reviewers highlighted our work as "novel, comprehensive, accessible" (ZEoJ, SRtx, eo9K) and the formalization of the task ambiguity problem as "fresh and solid" (SRtx, eo9K), with "valuable theoretical and practical contributions" (eo9K).

The main discussion points centered on improving the paper's clarity and further validating the performance of DAIL. We have addressed these points thoroughly in our rebuttal and will integrate the following improvements into the revised version:

1. **Expanded Baselines**: Following reviewers' suggestions, we have benchmarked DAIL against additional SOTA offline RL and meta-RL methods (TD3+BC, FOCAL, etc.). These new results demonstrate DAIL's substantial gains in handling task ambiguity.

2. **Theoretical Clarity**: Reviewers raised concerns about the clarity of definitions and theorem proofs. We have clarified and revised all unclear content, fully addressing their concerns.

3. **Sample Efficiency Analysis**: As suggested, we have supplemented our experiments with Area Under the Curve metrics and conducted Kolmogorov-Smirnov tests. The results confirm that our proposed components yield significant gains in sample efficiency.

4. **Quantitative Clustering Analysis**: In response to inquiries about the language embedding clusters, we have added Silhouette scores, which quantitatively show that our proposed enhancements significantly improve the discriminative power of the task embeddings.

We are pleased that Reviewers SRtx, ZEoJ, and eo9K were satisfied with our responses. To clarify Reviewer gNoU's remaining point, we would like to offer one final clarification on the practical impact of our theory.

1. **Improvement margin**: Reviewer gNoU raised a concern about whether our theoretical results translate to meaningful sample efficiency gains. Our theoretical analysis **provably** demonstrates that the distributional component in DAIL improves sample complexity, which is further validated by our empirical results, consistently showing significant gains of this component in sample efficiency in practice.

We sincerely thank the reviewers for their thoughtful and positive feedback. We also sincerely thank AC for the time and effort in leading the discussions.

Best,

The Authors

---

### Decision · Program_Chairs · 2025-09-17

**Decision:**

Accept (poster)

**Comment:**

The authors study the problem of language-conditioned (offline) RL tasks, identifying and rigorously examining the challenge of ambiguity of tasks in language instruction learning and their interpretation in relation to the final goal. Methodologically, they build on existing RL algorithms (e.g., conservative Q-learning (CQL)) from two perspectives: 1) from a policy learning perspective, they prove that distributional RL methods can improve the agent's ability to distinguish tasks and 2) from a representation learning perspective, they use 'trajectory-wise alignment' to optimize task representations through their correspondence to the language instructions (i.e., mutual information). The resulting algorithm, Distributional Aligned Learning (DAIL), is empirically evaluated on structured observations (BabyAI) and visual navigation (ALFRED) benchmarks to validate performance improvements over imitation learning (GCBC, BC-Z, GRIF) and offline RL algorithms (IQL, CQL) -- all adapted for language-conditioned settings. In summary, DAIL shows improvements in terms of success rate for BabyAI and path-length weighted success scores for ALFRED. Additionally, visualizations and ablation studies are also performed to better understand the dynamics of DAIL.

Strengths identified by the reviewers include:
- Task ambiguity is an important problem in contextual RL in general and language-conditioned RL specifically given recent increased interest in language representations (e.g., LLMs).
- The authors rigorously characterize the task ambiguity on language-conditioned tasks and theoretically analyze the proposed methods regarding distributional language-guided policy learning and trajectory-wise sentence alignment.
- The theory emits a practical implementation that is empirically evaluated against relevant baseline methods on challenging datasets -- and shown to perform well.
- The paper is well-written, the proposed method is conceptually appealing, and they key points are well-explained.

Weaknesses identified by the reviewers include:
- Reviewer gNoU had concerns regarding sufficient contextualization with respect to existing work, specifically contextual MDP and contrastive task representations. However, this was well-addressed in rebuttal both conceptually (i.e., pointing out distinctions) and empirically. Some of this should make its way into the main body of the paper and some to an appendix.
- Multiple reviewers had concerns regarding the significance of the methodological novelty (e.g., distributed RL, contrastive learning of similarity functions). While there is some merit to these concerns, the problem being solved, motivation, analysis, etc. are different (i.e., it is not rare to apply an existing solution once the problem is well-understood).
- Reviewer SRtx had concerns regarding sufficiency of the experiments in terms of comparing to state-of-the-art models (e.g., MCQ, TD3+BC, CCM, FOCAL, CSRO, UNICORN). These were adequately addressed in rebuttal -- although it would obviously be nice to see this integrated into the main text/appendices to expand the discussion more.
- There were multiple technical details from reviewers (e.g., eo9K, gNoU) that were addressed during rebuttal. Some of this expanded discussion, etc. should be integrated into the paper.

In summary, I believe the reviewer consensus is that the task ambiguity formulation for language-conditioned RL is an interesting perspective that leads to a practical algorithm that is well-analyzed (especially after rebuttal) for a timely and important problem. The experimental results are sufficiently convincing and the additional results provided during rebuttal strengthen the claims and can easily be integrated into the existing manuscript. Of particular note is that the rebuttal addressed all major concerns (in my assessment), even if not always noted by the reviewers. That being said, many of the weaknesses addressed still need to be integrated into the paper before it is ready for publication.